

# CH$_4$ emission estimates from an active landfill site inferred from a combined approach of CFD modelling and in situ FTIR measurements

Hannah Sonderfeld[1], Hartmut Bösch[1,2], Antoine P.R. Jeanjean[1], Stuart N. Riddick[3], Grant Allen[4], Sébastien Ars[5], Stewart Davies[6], Neil Harris[7], Neil Humpage[1], Roland Leigh[1], and Joseph Pitt[4]

[1]Earth Observation Science Group, Department of Physics and Astronomy, University of Leicester, Leicester, UK.
[2]National Centre for Earth Observation, University of Leicester, Leicester, UK
[3]Department of Civil and Environmental Engineering, Princeton University, NJ, United States
[4]Centre for Atmospheric Science, The University of Manchester, Manchester, UK.
[5]Laboratoire des Sciences du Climat et de l'Environnement (LSCE/IPSL), CNRS-CEA-UVSQ, Université de Paris-Saclay, Gif-sur-Yvette, France
[6]Viridor Waste Management Limited, Peninsula House, Rydon Lane, Exeter, Devon, UK
[7]Centre for Atmospheric Informatics and Emissions Technology, Cranfield University, Cranfield, UK

*Correspondence to:* Hannah Sonderfeld (hs287@le.ac.uk)

**Abstract.**

Globally, the waste sector contributes to nearly a fifth of anthropogenic methane emitted to the atmosphere and is the second largest source of methane in the UK. In recent years great improvements to reduce those emissions have been achieved by installation of methane recovery systems at landfill sites and subsequently methane emissions reported in national emission

inventories have been reduced. Nevertheless, methane emissions of landfills remain uncertain and quantification of emission fluxes is essential to verify reported emission inventories and to monitor changes in emissions. Here we present a new approach for methane emission quantification from a complex source like a landfill site by applying a Computational Fluid Dynamics (CFD) model to calibrated in situ measurements of methane as part of a field campaign at a landfill site near Ipswich, UK, in August 2014. The methane distribution for different meteorological scenarios is calculated with the CFD model and compared

to methane mole fractions measured by an in situ Fourier Transform Infrared (FTIR) spectrometer downwind of the prevailing wind direction. Assuming emissions only from the active site, a mean daytime flux of 0.83 mg m$^{-2}$ s$^{-1}$, corresponding to 53.3 kg h$^{-1}$, was estimated. The addition of a secondary source area adjacent to the active site, where some methane hotspots were observed, improved the agreement between the simulated and measured methane distribution. As a result, the flux from the active site was reduced slightly to 0.71 mg m$^{-2}$ s$^{-1}$ (45.6 kg h$^{-1}$), at the same time an additional flux of 0.32 mg m$^{-2}$ s$^{-1}$

(30.4 kg h$^{-1}$) was found from the secondary source area. This highlights the capability of our method to distinguish between different emission areas of the landfill site, which can provide more detailed information about emission source apportionment compared to other methods deriving bulk emissions.





## 1 Introduction

Methane ($CH_4$) is the second most important anthropogenic greenhouse gas (GHG) after carbon dioxide ($CO_2$) with a global warming potential of 34 on a 100 year time scale (Myhre et al., 2013). Globally, the $CH_4$ budget is reasonably well known, but on local and regional scales large uncertainties remain for emissions from individual sources (Dlugokencky et al., 2011).

The Climate Change Act 2008 legally binds the UK to reduce carbon emissions from GHG by 80 % in 2050 compared to the 1990 baseline (http://www.legislation.gov.uk/ukpga/2008/27/section/1), therefore a profound knowledge of $CH_4$ sources and their emission strength is required. The waste management sector contributed 3.7 % to total UK greenhouse gas emissions in 2014 (Brown et al., 2016) and is the second largest source of $CH_4$ in the UK after agriculture (Salisbury et al., 2016).

$CH_4$ and $CO_2$ are produced during the degradation process of municipal solid waste (MSW) at landfill sites. Under anaerobic
conditions landfill gas (LFG) with approximately 50 % $CH_4$ and 45 % $CO_2$ is produced (Czepiel et al., 1996). The organic degradable waste is broken down in several steps by initially aerobic and eventually anaerobic bacteria. While $CH_4$ is formed in the final steps from acetic acid decarboxylation or reduction of $CO_2$, $CO_2$ is formed in all stages (Czepiel et al., 1996; Themelis and Ulloa, 2007) of waste degradation. Once produced there are several ways for $CH_4$ to be released from the landfill site. It can be released through the landfill cover, where it partially oxidises to $CO_2$ depending on the cover soil, or
migrate underground and finally travel to the surface outside the landfill area (Scheutz et al., 2009). If a LFG recovery system is installed, the recovered $CH_4$ is either used for energy production or flared and thereby converted to $CO_2$. Modern gas recovery systems may reach efficiencies of over 90 % (Scheutz et al. (2009) and references therein).

The focus in past studies is on $CH_4$ emissions from closed and covered areas of landfills. Wide ranges of emissions are reported, which depend on the conditions of the site and cover. In years 1988 to 1994, Bogner et al. (1995) measured $CH_4$
fluxes in the range of -0.00154 to 1119 $\mathrm{g\,m^{-2}\,d^{-1}}$ at landfill sites in the US with different soil covers and with and without a LFG recovery system. Mønster et al. (2015) and Gonzalez-Valencia et al. (2016) report $CH_4$ fluxes in the range of 0.7 to 13.2 $\mathrm{g\,m^{-2}\,d^{-1}}$ from 15 Danish landfill sites and 10 to 575 $\mathrm{g\,m^{-2}\,d^{-1}}$ from three landfill sites in Mexico, respectively. One critical factor here is the installation and efficiency of a LFG recovery system (Bergamaschi et al., 1998).

Some studies also have analysed emissions from still operating landfill sites. Bergamaschi et al. (1998) reports a $CH_4$ flux of
up to 28.8 $\mathrm{g\,m^{-2}\,d^{-1}}$ for the uncovered area of a landfill site in Germany. At most landfill sites so called hotspots, e.g. cracks and leaks in the cover, are present, which emit much higher concentrations than the surrounding areas and have a high temporal variability (Rachor et al., 2013). To reduce uncertainty in landfill site emissions and the under representation of emissions from operating areas further accurate observations are needed.

A variety of techniques have been applied to quantify emissions from landfill sites in different stages. As a result of their
simplicity, chamber measurements are commonly used (Bogner et al., 1995; Czepiel et al., 1996; Börjesson et al., 2000; Christophersen et al., 2001; Schroth et al., 2012; Rachor et al., 2013). For this method static or dynamic flux chambers are placed in different locations on the landfill site and are sealed to avoid air exchange with the atmosphere. The increase in concentration of the target gas inside the enclosure is monitored. The main drawback of this technique is the sparse sampling of the area covered by the chambers. Inhomogeneity in emissions over a landfill site, e.g. caused by hotspots, can give misleading





results when scaling up to the whole landfill site. To overcome these difficulties a grid pattern is often chosen for placement of the chambers (Czepiel et al., 1996; Börjesson et al., 2000). Gonzalez-Valencia et al. (2016) recently tested a surface probe method for faster sampling of $CH_4$ emissions on discrete grid points by sampling in direct contact with the ground.

Eddy covariance (EC) systems also have been applied to measure nitrous oxide ($N_2O$) and $CH_4$ fluxes over landfill sites covering a wider area then enclosure techniques (Rinne et al., 2005; Lohila et al., 2007; Schroth et al., 2012). Although a good agreement to chamber measurements was found, this technique is dependent on the wind direction and sufficient wind speed (Lohila et al., 2007). They are best suited for flat terrain and have difficulties with complex topography.

In recent years tracer dispersion methods were developed and became more widely used (Czepiel et al., 1996; Galle et al., 2001; Foster-Wittig et al., 2015; Mønster et al., 2015). In this approach a tracer is released at the source and sampled downwind together with the target gas. Initially, sulfur hexafluoride ($SF_6$) (Czepiel et al., 1996) and $N_2O$ (Galle et al., 2001) were used as tracer, which are greenhouse gases themselves. Mønster et al. (2014) and Foster-Wittig et al. (2015) used acetylene as a tracer, which was co-measured with $CH_4$ with cavity ring-down spectroscopy (CRDS). This technique provides accurate measurements of $CH_4$ emissions of landfills and can also be applied to divide between several sources in one area by using an additional tracer (Scheutz et al., 2011; Mønster et al., 2014). A requirement for this method is accessibility downwind of the site for sampling the plume and the time span that can be covered is limited.

Atmospheric dispersion models appear as a useful tool for investigation of landfill site emissions. Delkash et al. (2016) used a forward model to analyse the effects of wind on short term variations in emissions in combination with a tracer method. Previously, Hrad et al. (2014) applied an inverse dispersion technique to emissions from an open windrow composting plant. They found an agreement of 10 to 30 % in an inter-comparison to tracer release experiments over five days.

The GAUGE (Greenhouse gAs Uk and Global Emissions) project aims for a better understanding and quantification of the UK GHG budget to support GHG emission reduction measures. In this context a two week field campaign between 4 and 15 August 2014 at a landfill site north of Ipswich, UK, was conducted as part of the GAUGE project to improve our understanding of landfill emissions and to investigate different methods for flux quantification. Here, we present simultaneous and continuous observation of $CO_2$ and $CH_4$ with in situ Fourier Transform Infrared (FTIR) spectroscopy at this landfill site. The application of a Computational Fluid Dynamics (CFD) model to the point measurements for estimating $CH_4$ fluxes is described and assessed. For complex terrains like a landfill site CFD models are expected to be more useful compared to Gaussian tools (Mazzoldi et al., 2008). This approach has the potential to provide a continuous data set for flux derivation from one set of CFD runs. It also offers the opportunity to identify and divide between different source areas.

In the following, the measurements during the field campaign are described and emission ratios are calculated initially to assess the influence of landfill emissions on the sampled air. Then the method for flux calculations with the CFD model outputs is presented. Emissions from the active site and a secondary source area are discussed.



## 2 Materials and methods

### 2.1 Experimental site

The landfill site under study is located in Great Blakenham near Ipswich (Fig. 1). In operation since 1992, it accepts a range of domestic and commercial/industrial waste and occupies approximately 330,000 $m^2$. The oldest part of the site, towards the

north is capped with a high-density polyethylene (HDPE) liner and covered with at least 1 m of restoration soils. East of the active area is a completed cell, which is temporarily capped with a HDPE only. The operational area (red area in Fig. 1) is located at a lower level to the centre of the site. Waste is deposited in this area on weekdays and Saturday mornings. The active waste is covered at the end of each day with a daily cover comprising soils and other inert materials. The site is equipped with an active gas control system comprising a network of gas extraction wells and associated pipework connected to four nominally

1 MWe LFG engines. Two high-temperature enclosed flares provide backup LFG control. All engines and flares are located in the gas utilisation plant (GUP) towards the southeastern end of the site.

    Measurements were carried out at different locations on the landfill site. With a focus on emissions from the active area, the main instrument used in this study (FTIR) was accommodated in a portakabin at the north end of the landfill site about 320 m downwind from there. Further instrumentation was located on the ridge above the active site, including meteorological

instruments and another greenhouse gas analyser to measure $CO_2$ and $CH_4$. This greenhouse gas analyser was either connected to a set of surface flux chambers or set up for sampling ambient air. A gas chromatograph (GC) for $CH_4$ measurements was installed at Inghams Farm approximately 700 m southwest of the landfill site. A cavity ring-down spectrometer measuring $CH_4$, $CO_2$, CO and $H_2O$ was located about 300 m northeast of the landfill on Chalk Hill Lane (Riddick et al., 2016).

### 2.2 Spectronus Trace Gas and Isotope Analyser

The instrument deployed at the northern edge of the landfill site in the portakabin was a Spectronus Trace Gas and Isotope Analyser by Ecotech (Knoxfield, Australia), further referred to as FTIR. Detailed descriptions of the FTIR can be found in Griffith et al. (2012) and Hammer et al. (2013). The built-in spectrometer is a Bruker IR cube with a range of 2000 to 7800 $cm^{-1}$ and a resolution of 1.0 $cm^{-1}$. The spectrometer measures the absorption of the air sample in a 3.5 L White cell. With a flow rate of 1 L $min^{-1}$ the standard sampling time of 3 min corresponds closely to a sample exchange in the cell. Before the

sample enters the cell it passes a Nafion dryer and a chemical dryer filled with magnesium perchlorate. Mole fractions of $CO_2$, $CH_4$, CO and $N_2O$, as well as the $^{13}CO_2$ isotopologue, are retrieved by software provided with the instrument. For this study we focus on the $CH_4$ measurements. Background spectra were recorded shortly before and during the campaign. A two point calibration was conducted on the last day of the measuring period with two primary standards of different mole fractions. They were calibrated at the Empa - Swiss Federal Laboratories for Materials Science and Technology, Dübendorf, Switzerland,

relative to the World Meteorological Organization (WMO) scale (WMO-$CH_4$-X2004A, WMO-$CO_2$-X2007, WMO-$N_2O$-X2006A, WMO-CO-X2014). For stability monitoring a target gas was measured daily. As no clear trend was observed with the target gas measurements no corrections were applied, but the observed variation was considered for estimation of the uncertainty. The combined uncertainty based on calibration with the primary gas standards and the target gas measurements




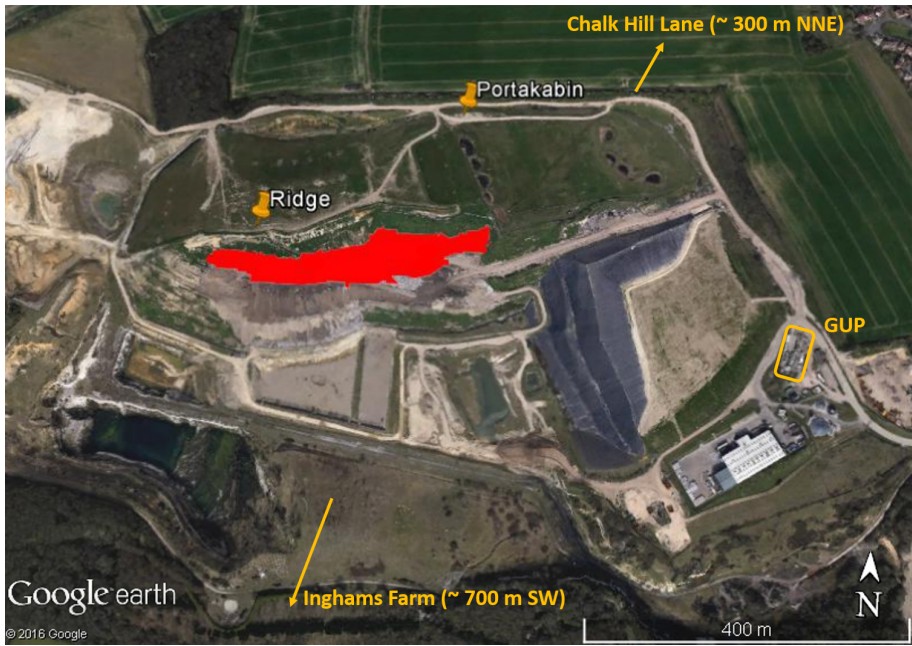

**Figure 1.** Birds view of the landfill site with the active site coloured in red in the centre. The portakabin with the FTIR is located at the north edge of the landfill site. Additional instrumentation was located at the ridge above the active site. A GC used for background measurements was situated about 700 m SW off site at Inghams Farm and a CRDS was operated on Chalk Hill Lane about 300 m NNE. The entry to the site with the weighbridge and the gas utilisation plant are at the east side.

is 0.44 ppm for $CO_2$ and 1.93 ppb for $CH_4$. The inlet for the FTIR was fixed to a tripod in front of the portakabin around 2 m above ground. Air was sampled through Teflon tubing with a flow of 1 L min$^{-1}$. A filter attached to the tubing prevented particles to enter the instrument. Irregularities in the power supply caused a delayed start of the measurements and another disruption later on. Additionally, a software error caused another gap in the data.

## 2.3 Background measurements and further instrumentation

To quantify the landfill $CH_4$ emissions, the background level of $CH_4$ needs to be distinguished from the enhancements in methane concentration related to the landfill emissions. Measurements by the University of Cambridge with a 200 series Ellutia GC-FID about 700 m off site to the southeast were used as background for southerly wind directions. For wind coming from the north, measurements of a Picarro cavity ring-down spectrometer, located northeast of the landfill site, are used as background. The set-up of both instruments is described in Riddick et al. (2016). Data were available with a time resolution of 15 min and uncertainty of 0.8 %. Additional measurements of $CO_2$ and $CH_4$ were taken occasionally at the ridge by the University of Manchester with an Ultraportable Greenhouse Gas Analyser (UGGA) by Los Gatos Research (Mountain View, California, USA), further referred to as UGGA, which is based on off-axis integrated-cavity output spectroscopy (Off-Axis ICOS). A detailed description of this technique can be found in Baer et al. (2002). An uncertainty of 1 % for the retrieved





mole fractions is stated by the manufacturer. This has been verified by subsequent laboratory calibrations, where the agreement between the UGGA and a WMO-traceable cylinder has been within this nominal uncertainty. Wind speed and direction were recorded at the ridge with a 3D sonic anemometer throughout the campaign.

### 2.4 CFD model

The gas dispersion from the landfill surface was calculated with a CFD model. CFD models use fluid dynamics equations constrained by boundary conditions that are solved numerically to calculate the behaviour of a fluid such as the wind within a particular domain (here the landfill terrain). CFD models require a complex parametrisation compared to traditional Gaussian dispersion models, but they have been shown to provide increased accuracy over complex terrain (Buccolieri and Sabatino, 2011), which can be considered to be the case over the landfill site.

The CFD simulations presented in this study have been validated previously by a comparison exercise against a wind tunnel experiment (Jeanjean et al., 2015). As a result of this comparison it was shown that a model accuracy of 30 % to 40 % can be achieved. This represents a slight amelioration in respect to traditional Gaussian dispersion modelling. The CFD simulations were performed under the OpenFOAM software platform. For calculating the wind flow, the Reynolds-averaged Navier-Stokes (RANS) k - $\epsilon$ model (Launder et al., 1975) was used. The dispersion of emissions from the landfill site was simulated with a
passive scalar transport equation (for full flow and boundary conditions see Jeanjean et al. (2015)).

A wall function was used to define the boundary conditions for the ground reproducing the landfill surface roughness. The landfill terrain was modelled with a roughness length value of 0.03 m, which corresponds to an open terrain with grass and a few isolated obstacles (WMO, 2008). A total number of 142 000 cells was used for this simulation. Boundaries for the mesh were set to (in British National Grid, minimum to maximum): X = [610350, 611650], Y = [249700, 250500], Z = [0, 500] with
initial cells of the domain assigned a dimension of 30 m. The cells corresponding to the terrain (ground) were assigned a size of 2 m and were kept constant up to 30 m away from the ground. Their resolution was then coarsened beyond 30 m with a maximum expansion ratio of 1.2. Topographic information for the CFD model were gained from a LIDAR (Light Detection And Ranging) survey of the landfill site. At its borders the LIDAR map was extended with a 5 m digital elevation model (Ordnance Survey).

### 25 3 Results and discussion

The landfill campaign took place between 4 and 15 August 2014. Initially, wind was coming from northeast with relatively low wind speeds (see Fig. 2, top panel). On 8 and 10 August wind came mainly from east to southeast, while the dominant wind direction on 9 and 11 to 12 August was from the south. At the end of the campaign the wind shifted more towards a westerly wind. The most frequent wind direction was around 210° (0°/360° corresponding to North) and wind speeds ranged from 0.1
to 13 m s$^{-1}$. The time series of measured $CH_4$ and $CO_2$ mole fractions are shown in Fig. 2 in the lower two panels colour coded with the wind direction. The active site lies roughly between 170° and 240° as seen from the portakabin. $CH_4$ values drop to background levels during measurements for air from the northern semi-circle (black and grey lines in Fig. 2), in the




$CO_2$ data a constantly low background value does not become apparent. High peaks in both gases appear before midnight on 8 August, when wind speeds were dropping to near zero, and in the following night for wind directions of 150° to 190°, which is only partially influenced by the active site. Two periods with wind constantly coming from the active area occurred during the course of the campaign: 9 August and 11 to 12 August. Air influenced by the active site was also measured during the night of 9 to 10 August until after midnight and on 14 August from the early morning hours to noon. These periods were less stable in wind direction compared to the former time periods.

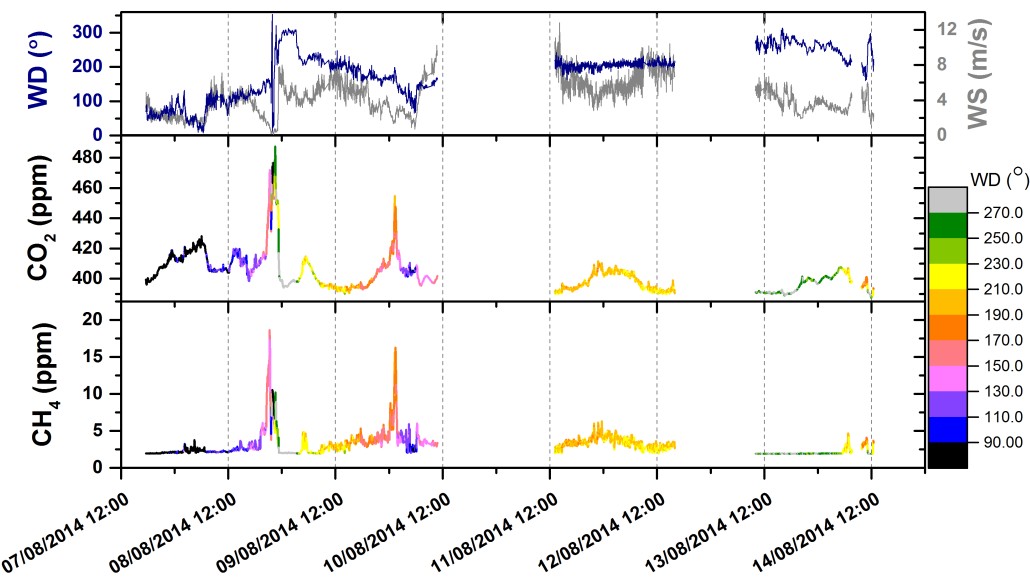

**Figure 2.** Time series of wind speed (WS, grey) and direction (WD, dark blue) in the top panel and of $CO_2$ and $CH_4$ colour coded with the wind direction. Black and grey refer to background air (270° to 90°), orange and yellow indicate air coming from the active site and blue to light pink and green colours mark transitional periods.

Much higher mole fractions with up to 700 ppm $CO_2$ and over 100 ppm $CH_4$ were observed by the UGGA at the ridge. These particularly high values were measured before the FTIR measurements were started, so a direct comparison here is not possible. Towards the end of the campaign both instruments were operated at the same time. Mole fractions measured then were much lower compared to the beginning, but values at the ridge were still enhanced compared to the portakabin. Chamber measurements along the south side of the ridge leading down to the active site showed that the cover of the old landfill part was not leak tight and allowed for additional significant emissions. $CH_4$ migrating underneath the landfill cap can leak out at places where the landfill cover is interrupted, e.g. at the edge of a side slope or through cracks in the cap. This is a common issue at landfill sites and highly variable emissions from these hotspots have been reported (Di Trapani et al., 2013; Rachor et al., 2013; Gonzalez-Valencia et al., 2016). Although they contribute to the total GHG emissions of a landfill, measurements within





the proximity of those hotspots are not suitable for estimation of the bulk emissions. High temporal variability and spatial inhomogeneity would result in non representative fluxes. Hence, the application of the CFD model to the ridge measurements is not presented here.

## 3.1 Emission ratios

The ratio of ppm $CH_4$ per ppm $CO_2$ at the location of the emission source is often referred to as emission ratio and is given here in $\mathrm{ppm\,ppm^{-1}}$ for simplicity. It can provide insights into the degree of $CH_4$ oxidation at landfill sites (Gebert et al., 2011; Pratt et al., 2013). Under anaerobic conditions the landfill gas is typically enriched in $CH_4$ and results in ratios of 1.2 to 1.5 $\mathrm{ppm\,ppm^{-1}}$ for $CH_4$ to $CO_2$ (Lohila et al., 2007; Gebert et al., 2011). On site continuous monitoring undertaken in a borehole by Ground-Gas Solutions (GGS) detected LFG ranging from 59 to 67 % $CH_4$ and 31 to 42 % $CO_2$, which results in a mean ratio of 1.8 $\mathrm{ppm\,ppm^{-1}}$. The FTIR at the portakabin measures the combined mixing ratio $\chi_{meas}$ of the background $\chi_{bg}$ and the emission from the landfill $\chi_{lf}$. Thus, we are looking at the enhancement mixing ratio ($\Delta CH_4 = \chi_{lf} = \chi_{meas} - \chi_{bg}$) of $CH_4$ divided by the enhancement of $CO_2$ ($\Delta CO_2$), which gives us the observed enhancement factor $EF = \Delta CH_4 / \Delta CO_2$ (Lefer et al., 1994). This corresponds to the emission ratio as long as there are no additional sources or sinks along the transport pathway.

The EF can also be directly determined by the slope from plotting $CH_4$ versus $CO_2$ without subtracting a background value beforehand (Yokelson et al., 2013). For $CO_2$, background measurements were sparse, hence the EF is determined from the correlation of $CH_4$ to $CO_2$ (Fig. 3). Data for periods influenced by the active site are plotted separately for day (9 am to 6 pm UTC) and nighttime (9 pm to 6 am UTC) as the background of $CH_4$ and $CO_2$ is expected to change during the course of a day. That way EF is derived from data with comparable background values. Data inbetween the day and nighttimes showed a gradual shift in background concentration, which leads to artificially lower EF.

Results for the EFs are given in Table 1. A similar slope was observed for all three days and the two nights. The EF are in the range of 0.16 to 0.27 $\mathrm{ppm\,ppm^{-1}}$ with a mean of $(0.23 \pm 0.04)\,\mathrm{ppm\,ppm^{-1}}$. There is a correlation in all cases with $R^2$ between 0.393 to 0.857. The lowest correlation coefficient was observed for 9 August 2014, when the wind was less stable and covered a wider range in wind directions then on the other days. Compared to air masses coming from the north $CH_4$ is enhanced, but the EF is significantly lower than would be expected from landfill gas from underneath the cover. This suggest that the sampled air during these phases had picked up emissions from the active site, which is enriched with $CH_4$ but due to the exposure to air is more oxidised than landfill gas.

Daytime EF measured at the ridge, closer to the active site, with the UGGA ranged from 0.42 to 0.54 $\mathrm{ppm\,ppm^{-1}}$. These ratios are still representative of waste degradation under aerobic conditions, but show a higher $CH_4$ content compared to the EF observed at the portakabin. Processes at the surface of a landfill site can alter the $CO_2$ concentration (Scheutz et al., 2009). Hence, interpretation of the EF as an estimate for the emission ratio with regard to the degree of $CH_4$ oxidation can be difficult. The difference can be explained by additional $CO_2$, which was taken up by the air masses during the transport over the capped area between the ridge and the portakabin. Closed chamber measurements by GGS found a $CO_2$ flux of $0.1587\,\mathrm{mg\,m^{-2}\,s^{-1}}$ in this area, but no significant $CH_4$ emissions.



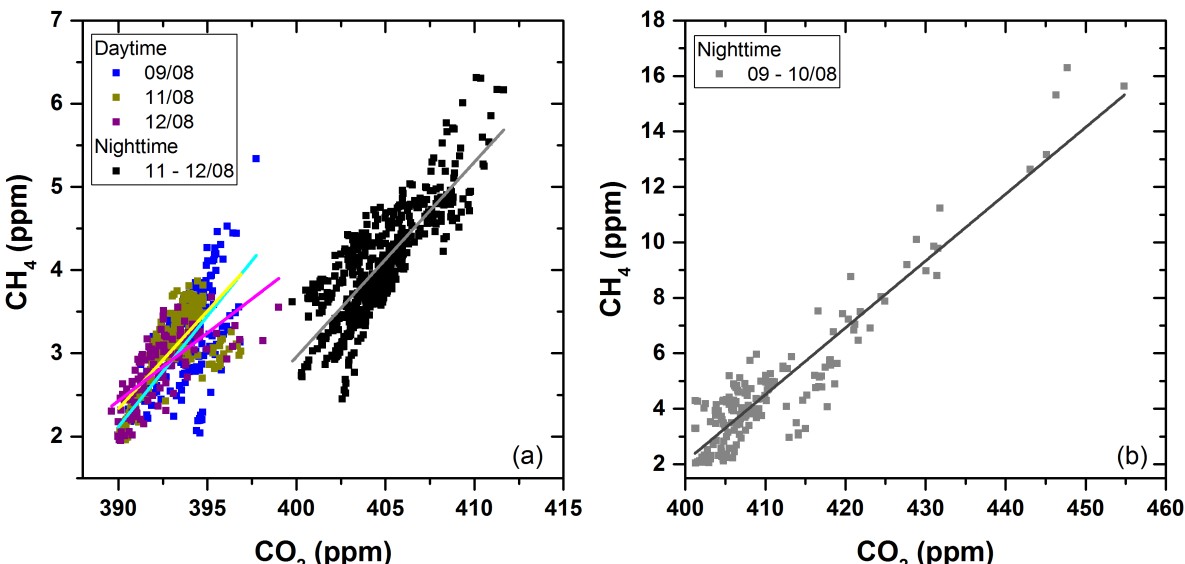

**Figure 3.** Determination of the enhancement factor as the gradient from the correlation of $CH_4$ to $CO_2$ separately for three days (9 am to 6 pm UTC) and two nights (9 pm to 6 am UTC) influenced by air from the active site. Data are shown in two separate graphs to account for the different scales.



**Table 1.** EF given as ppm $CH_4$ per ppm $CO_2$ with fit uncertainty and $R^2$ as determined from the slope of the correlation of $CH_4$ to $CO_2$ measured at the portakabin for day (9am to 6 pm UTC) and nighttime (9 pm to 6 am UTC) separately.

| Date | Day/Night | EF (ppm ppm$^{-1}$) | $R^2$ |
|---|---|---|---|
| 09/08 | Day | $0.266 \pm 0.026$ | 0.393 |
| 11/08 | Day | $0.235 \pm 0.012$ | 0.572 |
| 12/08 | Day | $0.163 \pm 0.015$ | 0.499 |
| 09 to 10/08 | Night | $0.241 \pm 0.007$ | 0.857 |
| 11 to 12/08 | Night | $0.234 \pm 0.007$ | 0.655 |





## 3.2 Methane distribution

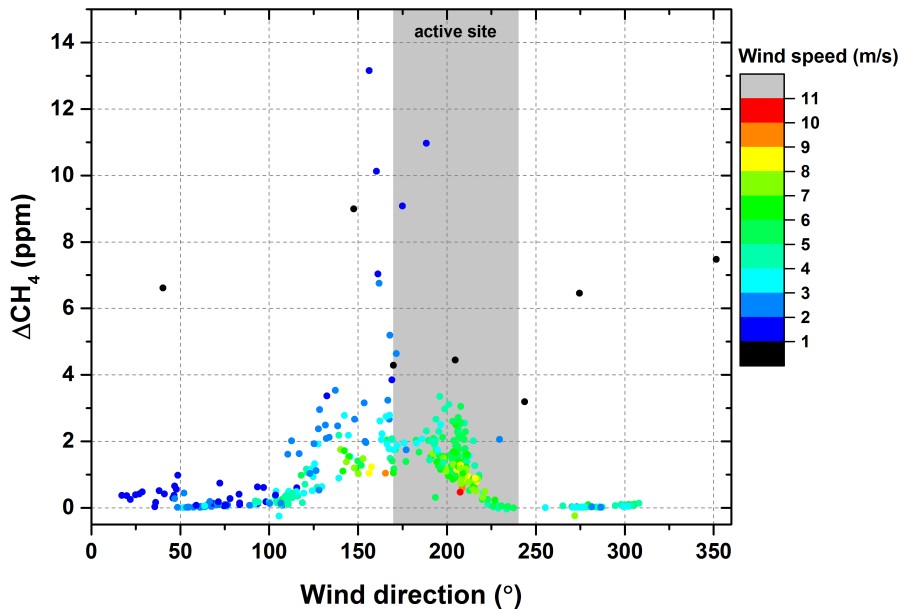

**Figure 4.** $CH_4$ distribution with wind direction and colour coded with the wind speed based on 15 min averages. The wind direction range of the active site is marked in grey.

The $CH_4$ concentration data were averaged over 15 min and the background $CH_4$ concentration was subtracted by using the GC data for wind directions from the south and the Picarro data for wind coming from the north to derive the excess mixing ratio $\Delta CH_4$. In the morning of 8 August the wind direction changed rapidly from around 20° to 100° and high $CH_4$ concentrations were observed with the GC resulting in negative values when subtracting the background from the FTIR data. Figure 4 shows the distribution of $\Delta CH_4$ with the wind direction. Between 120° and 220° $CH_4$ levels are clearly elevated when wind is passing the landfill site before reaching the portakabin. Highest concentrations are observed during low wind speeds when emissions accumulate. Generally the wind speed was higher for wind directions above 150°. Two maxima at around 140° and 200° stand out. The maximum at 140° is from air passing the GUP close to the weighbridge of the landfill site and the fully filled but not yet fully restored area. Further on we focus on the elevated $CH_4$ level at around 200° where emissions from the active site have been picked up.





### 3.3 Application of CFD model to the in situ data for flux calculations

The CFD model is applied to simulate the distribution of $CH_4$ concentrations emitted from the active site of the landfill at the point of measurement for different meteorological scenarios. Over the estimated area of the active site of $A = 17,823 \, \mathrm{m^2}$ (encircled area in Fig. 1) a constant emission $f_{Source}$ normalised to $1 \, \mathrm{g \, s^{-1}}$ is set. Figure 5 (left side) shows a 1 m grid resolved topographic map from the LIDAR survey of the landfill site. The red area in the topographic map marks the active site of the landfill site. Figure 5 (right side) shows the simulated concentration of the emitted compounds by the CFD model for the position of the portakabin at 2 m height depending on the wind direction for four different wind speeds. The units used by the CFD model correspond to a mass concentration $\rho_{Source}$ in $\mathrm{g \, m^{-3}}$ which is converted to mole fractions $\chi_{Source}$ for $CH_4$ with a molar mass of $M_{CH_4} = 16.04 \, \mathrm{g \, mol^{-1}}$ for comparison with the measurements (Eq. 1). The molar concentration of air $c_{Air}$ is $40.34 \, \mathrm{mol \, m^{-3}}$.

$$\chi_{Source} = \frac{\rho_{Source}}{c_{Air} \cdot M_{CH_4}} \cdot 10^6 \, \mathrm{ppm} \tag{1}$$

The ratio of measured, $\chi_{FTIR}$, to modelled mole fraction is used to scale the normalised emission and calculate the $CH_4$ flux with Eq. (2).

$$F_{CH_4} = \frac{f_{Source} \cdot \chi_{FTIR}}{A \cdot \chi_{Source}} \tag{2}$$

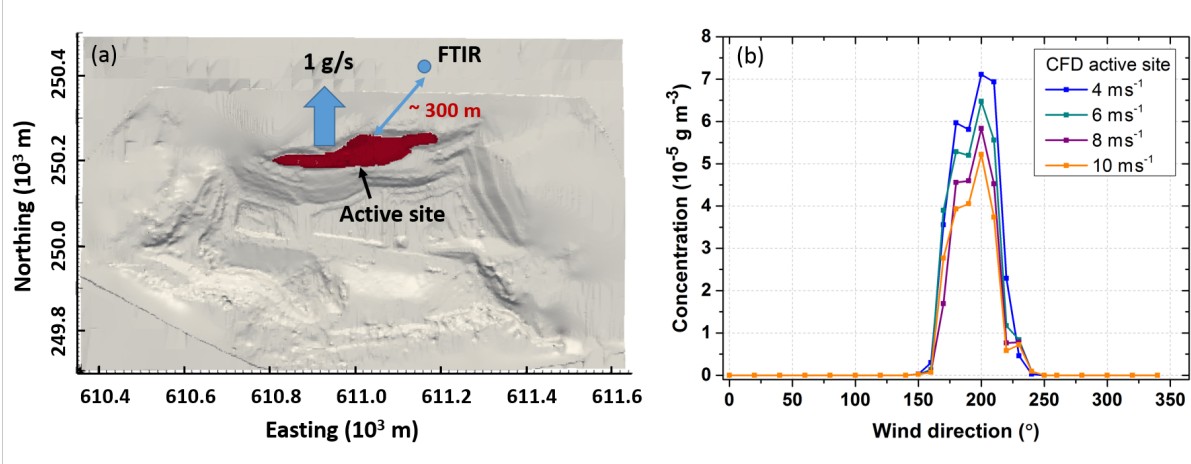

**Figure 5.** The emission area used for the CFD approach is marked in red on the topographic map (a). The results of the CFD model for the position of the FTIR measurement site are shown in (b).





The CFD model calculates only the enhancement above background in concentration based on the defined emissions $f_{Source}$. Therefore, the outputs correspond to the excess mixing ratios $\Delta CH_4$ in Eq. (2). As before for the enhancement ratios, data were analysed separately for day and night for periods with air mainly coming from the active site (daytime: 9, 11 and 12 August and nighttime: 11 to 12 August). The background values were calculated from the off site GC measurements as the

mean over these periods and are given in Table 2.

The CFD model results refer to wind speeds (WS) of 4, 6, 8 and 10 m s$^{-1}$ for all wind directions (WD) and are given as WS/WD pairs of 10° between 140° and 260° and 20° elsewhere. Taking this into account, the mean of the FTIR data was calculated around these model output pairs. Mean mole fractions and their standard deviations of $CH_4$ with at least 5 data points per bin are shown in Fig. 6. High $CH_4$ concentrations are observed in the range of 170° to 200°, decreasing towards

more westerly wind directions. On these days no significant amount of data for wind directions below 170° was collected. Especially on 11 August and during the night 11 to 12 August a distinction of the data based on the wind speed with higher values for lower wind speeds can be seen. For each of the analysed days and the single night the mean background for $CH_4$ was calculated from the GC data in the same time period. Table 2 summarises the mean fluxes and their standard deviations with the respective background values. The given uncertainty in percent refers to the combined error of the standard deviation of

$\Delta CH_4$ and a model uncertainty of 40 %, which is the bigger contribution. Measurement uncertainties are significantly smaller and were not taken into account here.

For the different days the calculated fluxes are in good agreement. For the night a higher flux was found. The calculated fluxes are given in Table 2 refer to wind directions below 220° only. The steep decline in concentration at 220° based on the CFD model results was not observed in the FTIR data. The fluxes inferred for this range are up to a factor of 6 higher. Additional

unknown sources, which were not taken into account by the CFD model could cause the enhanced $CH_4$ concentrations from this direction.

Instead of calculating the flux for each WS/WD pair separately, the CFD outputs were also fitted to the FTIR data with a linear least square fit over all wind directions present for each day/night and wind speed using Eq. (3).

$$\chi_{FTIR,i} = \frac{f_{Source} \cdot F_{CH_4}}{A} \cdot \chi_{Source,i} \qquad (3)$$

A robust fitting method using an M-estimator to reduce the influence of outliers was also tested, but did not have a significant effect on the results. Hence, only the results from the linear least square fit are reported in the following (Table 3). The standard errors are the fit uncertainty of the coefficient. Inferred fluxes range from 0.66 to 0.92 mg m$^{-2}$ s$^{-1}$ during daytime and 1.37 to 1.39 mg m$^{-2}$ s$^{-1}$ at night. When all daytime data are fitted together an overall flux of $(0.83 \pm 0.04)$ mg m$^{-2}$ s$^{-1}$ is obtained. This results in $CH_4$ emissions of 53.3 kg h$^{-1}$ over the active site. It should be noted that the CFD model was validated against

bag samples in a tracer release experiment at the landfill site and turbulence mixing parameters were optimised to match the bag samples. Hence, the CFD outputs correspond to daytime conditions and fluxes calculated for nighttime need to be used with care, but are included here for completeness. Generally, it can not be predicted how the CFD output would change with




decreased turbulence, as it would be the case during night, as it highly depends on the location of the measurement and the meteorological conditions.

Enhancements in $CH_4$ ($\Delta CH_4$) simulated from the inferred fluxes (Table 3) are shown in Fig. 7 together with the in situ data. Around 200° the measurements are well represented by the model, but model estimates were found to be lower for other

5     wind directions. This is mainly the case for low wind speeds, where more $CH_4$ can accumulate, and wind directions further south east.

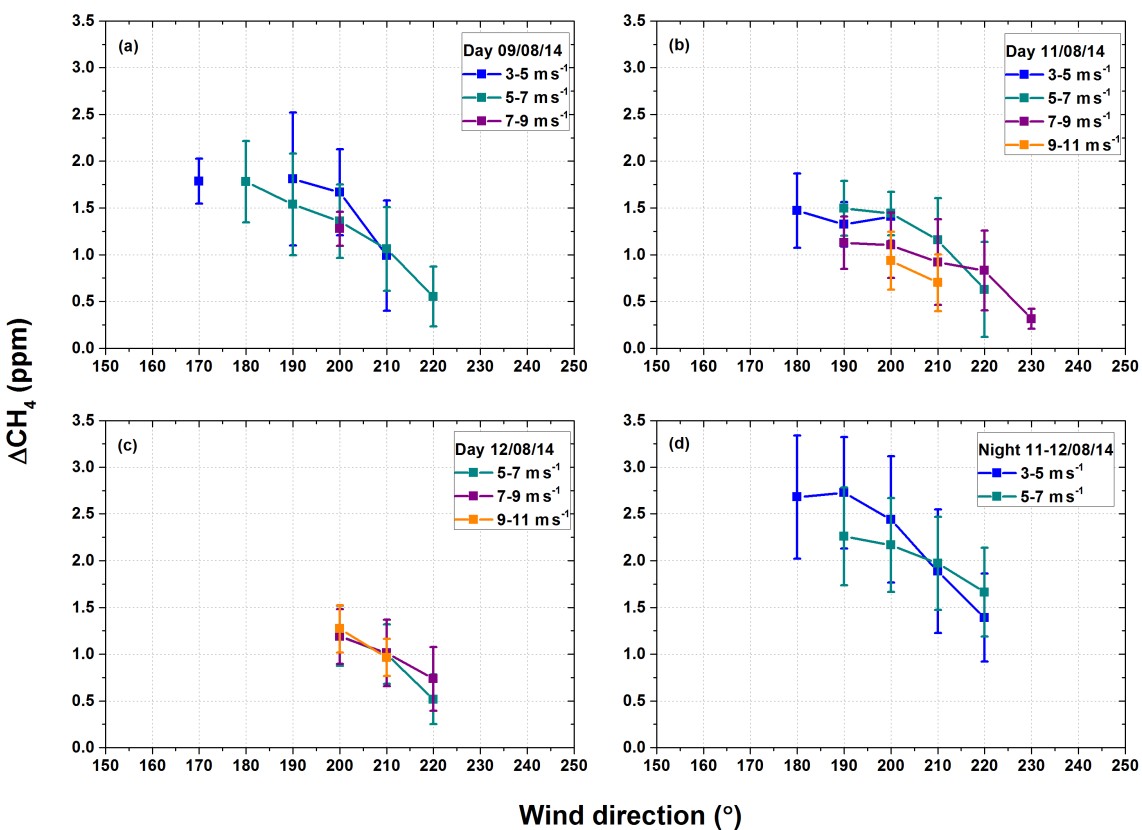

**Figure 6.** $\Delta CH_4$ averaged bin wise matching the CFD outputs for each day ((a) - (c)) and the one night (d) with wind coming from the active site. The standard deviation is plotted as error bars. Data are only shown for more then five data points per bin.





**Table 2.** Mean emission fluxes and standard deviations of $CH_4$ calculated from the FTIR data and CFD results together with the respective background value (BG). The uncertainty range is derived from error propagation based on the standard deviation of $\Delta CH_4$ and the model uncertainty.

| Date | Day/Night | BG (ppm) | Flux (mg m$^{-2}$ s$^{-1}$) | Uncert. Flux (%) |
|---|---|---|---|---|
| 09/08 | Day | 1.898 | $0.99 \pm 0.39$ | 40.4 - 44.9 |
| 11/08 | Day | 1.869 | $0.79 \pm 0.12$ | 40.6 - 43.2 |
| 12/08 | Day | 1.867 | $0.78 \pm 0.11$ | 40.6 - 41.9 |
| 11 to 12/08 | Night | 1.911 | $1.38 \pm 0.26$ | 41.8 - 43.6 |





**Table 3.** Results of a linear least square fit of the CFD model to the in situ data. $CH_4$ fluxes were fitted for each day/night and wind speeds separately. The standard error for the flux, adjusted $R^2$, the residual standard error (RSE) and degrees of freedom (df) are also shown.

| Date | WS (m s$^{-1}$) | CH$_4$ Flux (mg m$^{-2}$ s$^{-1}$) | Stand. Error (mg m$^{-2}$ s$^{-1}$) | Adj. R$^2$ | RSE (ppm) | df |
|---|---|---|---|---|---|---|
| **Day 09/08** | 4 | 0.89 | 0.22 | 0.805 | 0.71 | 3 |
| | 6 | 0.92 | 0.11 | 0.928 | 0.36 | 4 |
| | 8 | 0.80 | | | | 0 |
| **Day 11/08** | 4 | 0.80 | 0.05 | 0.987 | 0.16 | 2 |
| | 6 | 0.87 | 0.10 | 0.950 | 0.27 | 3 |
| | 8 | 0.79 | 0.15 | 0.845 | 0.36 | 4 |
| | 10 | 0.66 | 0.01 | 0.999 | 0.02 | 1 |
| **Day 12/08** | 6 | 0.68 | 0.09 | 0.950 | 0.21 | 2 |
| | 8 | 0.80 | 0.20 | 0.833 | 0.41 | 2 |
| | 10 | 0.90 | 0.02 | 0.999 | 0.04 | 1 |
| **Night 11 to 12/08** | 4 | 1.37 | 0.16 | 0.936 | 0.58 | 4 |
| | 6 | 1.39 | 0.27 | 0.865 | 0.75 | 3 |





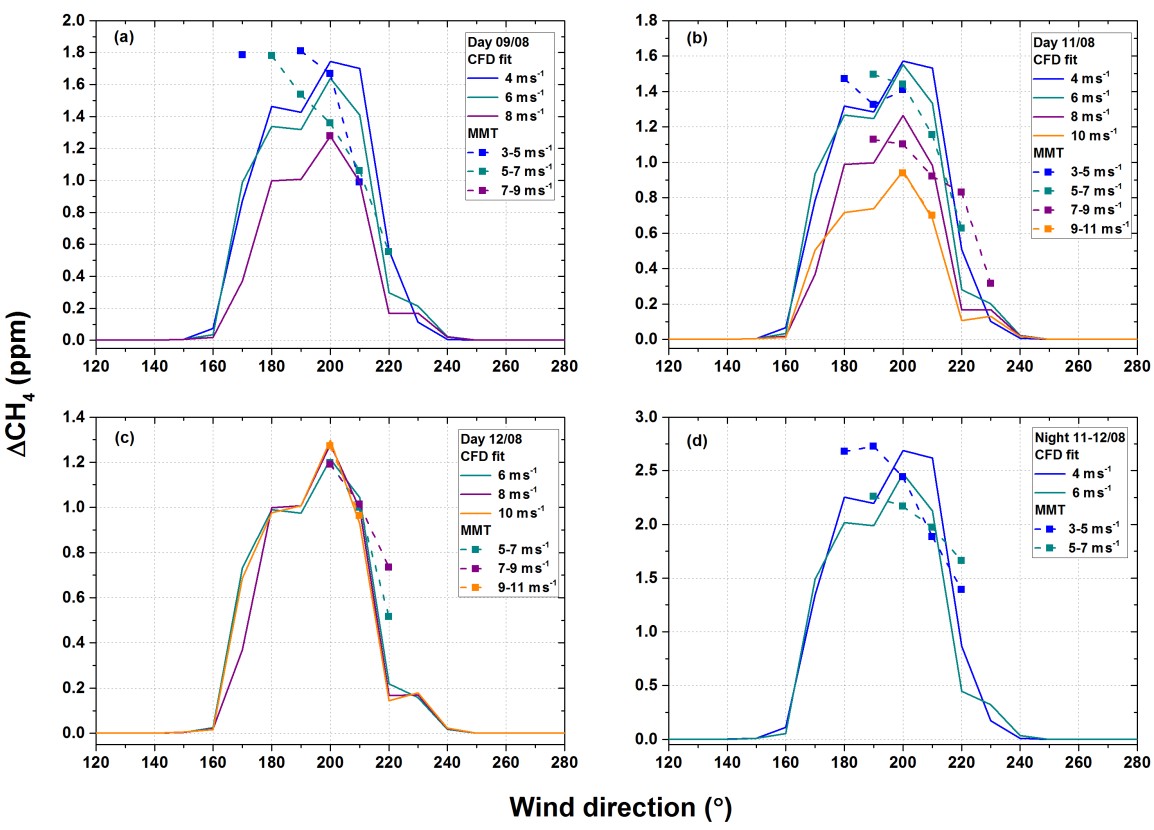

**Figure 7.** Measured (MMT) and simulated (CFD fit) $\Delta CH_4$ based on linear fit of the CFD model to the FTIR data.

## 3.4 Inclusion of an additional source area

As described in the previous section, the CFD model results in a steep decline in simulated $CH_4$ concentration at wind directions of 220° and further west, while measurements are still enhanced. No $CH_4$ emissions were observed on top of the restored section of the landfill site between the ridge and the portakabin, but emission hotspots were detected on the south side of the

5   ridge above the active site, further referred to as side area (see red area in Fig. 8). Thus, we have included a secondary source area $A_{side}$ in our analysis estimated to be 26,400 $m^2$. Gaps in the top liner along the side allow for $CH_4$ to escape underneath a soil cover with some vegetation. These emissions are directly adjacent to the emissions from the active site and are thereby also detected by the FTIR for wind coming from the south to southwest. The emission strength compared to the active site is unknown and can be expected to be highly variable (Rachor et al., 2013). To take these into account, a second CFD run for the



described area as emission source was set up. For a normalised source flux of $f_{source} = 1\,\mathrm{g\,s^{-1}}$ concentration distributions as shown in Fig. 8 (right side) are modelled.

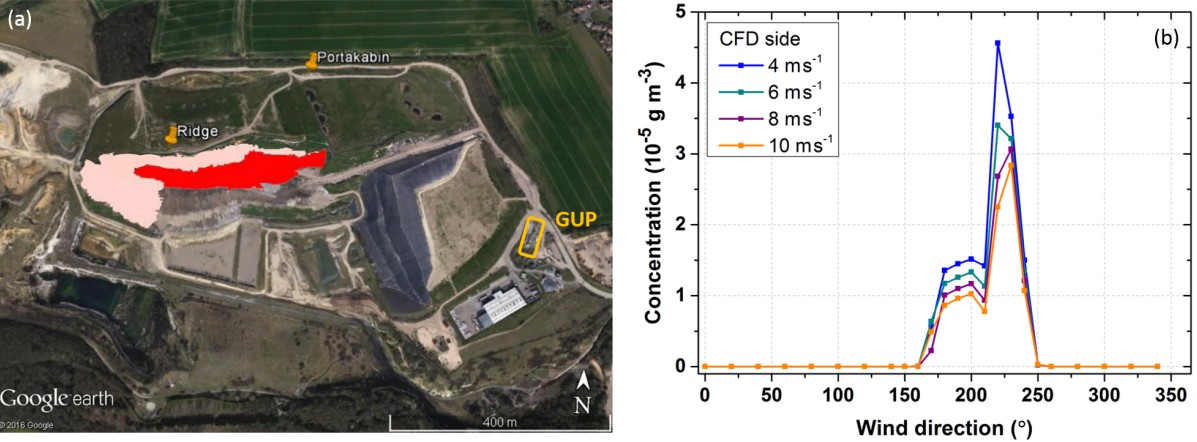

**Figure 8.** (a) Secondary source area (lightpink) between the active site (red) and the ridge and (b) CFD modelled concentration for the secondary source area only.

Flesch et al. (2009) discussed the requirement of having two sensors in different places for a two source problem. But they also describe the possibility of solving the problem with a single sensor, if the range in meteorological conditions is broad

enough. Here, we have only one sensor available, but a range in wind speeds and direction for most days. The modelled concentrations were combined with Eq. (4) to calculate the fluxes from both areas under assumption that the measured $CH_4$ concentration is an accumulated signal of the emissions from the active site and the side.

$$\chi_{FTIR,i} = \frac{A_{active} \cdot \chi_{active,i}}{f_{source}} F_{active,i} + \frac{A_{side} \cdot \chi_{side,i}}{f_{source}} F_{side,i} \qquad (4)$$

Equation (4) was applied in two ways. First, a linear least square fit was applied to the data of each day and night separately

for each wind speed. Secondly, all daytime data were fitted with a linear least square together to derive a mean flux. Fluxes from both source areas for each set of data are given in Table 4 together with their fit uncertainty as standard error and the residual standard error. The same robust fitting methods were applied again to take outliers into account. The results were found to be consistent with each other within the fit uncertainty. Hence, only results from the linear least square fit are reported.

The combined fit in cases where data are only available for the lower wind directions, such as for $4\,\mathrm{m\,s^{-1}}$ on 9 and 11 August

2014, does not result in realistic coefficients for the fluxes and in conjunction with their large errors can not be considered as representative values. For wind speeds of $10\,\mathrm{m\,s^{-1}}$ only two data points were available, i.e. zero degrees of freedom, and the fit assigned a much higher flux to the side area and only a minor contribution to the active site. Therefore these fits were not further included.





Figure 9 shows simulated $\Delta CH_4$ derived based on fluxes calculated from the separate fits with combined CFD runs in comparison to the measurements. Results given in brackets in Table 4 are not shown. At the peak wind direction both approaches show similar good agreement between the model and the measurements. Measured $CH_4$ concentrations at 220° are much better represented by the combined CFD model compared to the model run based on the active site only (Fig. 9). The mean residual standard error (RSE) could be reduced from 0.42 to 0.25 ppm based on equivalent fits from 9 August (6 m s$^{-1}$), 11 August (6 and 8 m s$^{-1}$), 12 August (6 and 8 m s$^{-1}$) and night of 11 to 12 August 2014 (4 and 6 m s$^{-1}$). The mean fluxes from the same daytime data combined in one fit are $(0.71 \pm 0.05)$ mg m$^{-2}$ s$^{-1}$ for the active site and $(0.32 \pm 0.08)$ mg m$^{-2}$ s$^{-1}$ for the side. From this the overall emissions are 76.0 kg h$^{-1}$ over an area of 44,223 m$^2$.





**Table 4.** Results of a linear least square fit of combination of the two CFD model outputs for the active site and the side to the in situ data. $CH_4$ fluxes were fitted for each day/night and wind speeds separately. The standard error for the flux, adjusted $R^2$, the residual standard error (RSE) and degrees of freedom (df) are also shown.

| Date | WS | Active site | | Side | | Adj. $R^2$ | RSE | df |
|---|---|---|---|---|---|---|---|---|
| | | $CH_4$ Flux | Stand. Error | $CH_4$ Flux | Stand. Error | | | |
| | (m s$^{-1}$) | (mg m$^{-2}$ s$^{-1}$) | | (mg m$^{-2}$ s$^{-1}$) | | | (ppm) | |
| **Day 09/08** | 6 | 0.84 | 0.16 | 0.22 | 0.30 | 0.919 | 0.38 | 3 |
| **Day 11/08** | 6 | 0.76 | 0.10 | 0.29 | 0.17 | 0.970 | 0.21 | 2 |
| | 8 | 0.65 | 0.13 | 0.36 | 0.17 | 0.919 | 0.26 | 3 |
| **Day 12/08** | 6 | 0.59 | 0.01 | 0.23 | 0.02 | 1.000 | 0.02 | 1 |
| | 8 | 0.60 | 0.04 | 0.56 | 0.06 | 0.996 | 0.07 | 1 |
| **Night 11 to 12/08** | 4 | 1.23 | 0.21 | 0.36 | 0.35 | 0.936 | 0.58 | 3 |
| | 6 | 1.03 | 0.12 | 0.97 | 0.19 | 0.985 | 0.25 | 2 |




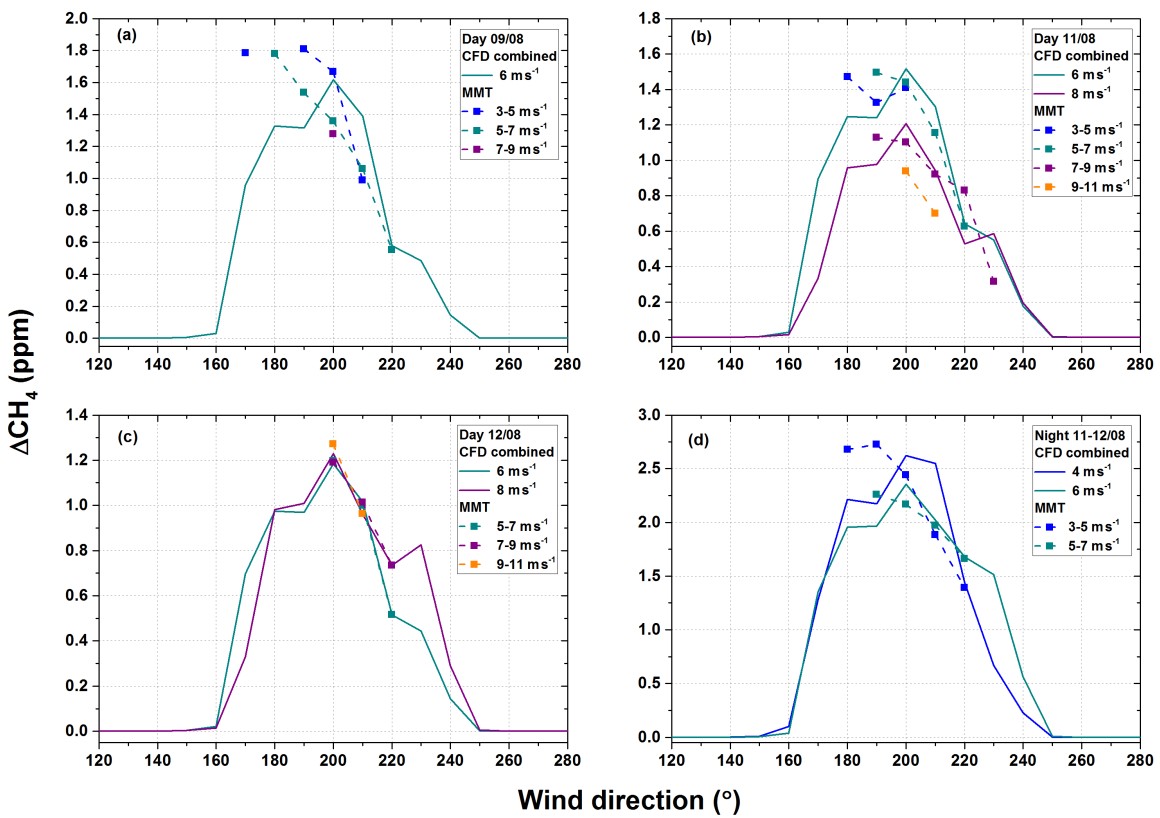

**Figure 9.** Measured (MMT) and simulated (CFD combined) $\Delta CH_4$ based on a linear fit combining the CFD model for the active site and the side.

### 3.5 Comparison to other flux estimations

Based on the CFD approach considering the active area, a mean daytime $CH_4$ flux of $0.83 \, \mathrm{mg \, m^{-2} \, s^{-1}}$ was calculated, which corresponds to $53.3 \, \mathrm{kg \, h^{-1}}$. Including emissions from the side area results in an overall flux of $76.0 \, \mathrm{kg \, h^{-1}}$ over a total area of $44,223 \, \mathrm{m^2}$. $CH_4$ fluxes from the landfill site were also measured by two other groups during the landfill campaign. Riddick et al. (2016) used an atmospheric inverse dispersion model to determine fluxes from the off site $CH_4$ measurements between July and September 2014. They assume an emitting open site of approximately $70,000 \, \mathrm{m^2}$. With $0.709 \, \mathrm{mg \, m^{-2} \, s^{-1}}$ on average over day and night they observed a $CH_4$ flux in good agreement to the one determined in this work. Based on the larger area the total flux in Riddick et al. (2016) corresponds to $178.7 \, \mathrm{kg \, h^{-1}}$. They report a similar uncertainty of 42 % to our approach. Mønster and Scheutz (2015) applied a dynamic tracer dispersion method to estimate total $CH_4$ emissions from the landfill





(total area: 330,000 $\mathrm{m}^2$) between 5 and 12 August 2014. They derived fluxes in the range of 217 to 410 $\mathrm{kg\ h}^{-1}$ with a standard error of 14 to 42 % from six experiments in this period. $CH_4$ emissions estimated by the landfill site's owner are around 2,230 tonnes in 2014, which corresponds to an annual mean flux of 254.6 $\mathrm{kg\ h}^{-1}$. This value is calculated from the total $CH_4$ as modelled based on waste input to the site and the LFG consumed by the power plant.

Compared to the other two methods we derived a lower $CH_4$ flux from the landfill site based on the on site measurements at the portakabin. The approaches of Riddick et al. (2016) and Mønster and Scheutz (2015) aim at quantifying the integrated signal of the whole landfill site, while our CFD approach focussed on emissions from the active site only (and separately the side area). Hence, fluxes obtained by these bulk emission methods are likely to be higher, including emissions from other areas, then the ones derived with the CFD approach. Indications for further emissions from wind directions towards the GUP and the

temporarily capped completed cell in the south east were visible in the $CH_4$ distribution measured with the FTIR (Fig. 4), but were not subject of the present study. Definition of the source area is a crucial part for setting up the CFD simulation and needs to be carefully assessed. When comparing fluxes inferred from different methods the source areas used need to be accounted for. As an advantage of the CFD approach, several source areas with different emission strength can be included.

## 4   Summary and conclusions

We presented a new approach to quantify $CH_4$ emissions from a defined source area at a landfill site. To this end, precise in situ measurements were combined with a CFD model. The CFD model only needs to be run once to cover the whole range in meteorological conditions and can then be applied to a series of continuous in situ measurements. Additionally, meteorological data and background measurements are needed for application of the CFD model. The measurements can be maintained over a longer period without great effort and can thereby cover a wide range of various environmental conditions.

Consistent fluxes from the active site were found for three different days with stable meteorological conditions. Data from wind directions of 220° were not well reproduced by the CFD outputs for the active site only. Taking emissions from the side area between the active site and the ridge into account improved the agreement between measurements and model in this area. This shows that the emission source in the CFD model needs to be well defined. This is challenging for a complex terrain like a landfill site, where several sources of $CH_4$ with different emission strength exist. Chamber measurements or an initial walk

over survey are valuable tools to characterise different parts of a landfill site and detect emissions which are otherwise easily overseen by point measurements. The main uncertainty results from the model accuracy.

With our approach we estimated $CH_4$ emissions between 53 and 76 $\mathrm{kg\ h}^{-1}$ by the active site and surrounding area, depending on the area taken into account with the CFD model. Compared to the total emission estimate from the landfill site's owner and the bulk emission approaches by Riddick et al. (2016) and Mønster and Scheutz (2015), this assigns a smaller contribution

to the active site and suggests additional significant $CH_4$ emissions from other parts of the site. Enhanced $CH_4$ was observed for wind directions further east of the active site (Fig. 4) where the CFD model does not show any contribution from the active site. The presented study shows that the CFD approach can be used to assess the emission strength from a well defined area in a terrain with several emission sources. Different source areas can be distinguished and considered for emission estimates.





For this reason, the instrument should be positioned in the proximity of the source area, but still in a distance great enough to detect its whole signal and reduce the influence of highly variable point sources such as landfill hotspots. For estimation of bulk emissions, off site measurements in a greater distance to the source are more useful.

## 5 Data availability

5    Data are available upon request from the authors.

*Acknowledgements.* We thank NERC for their funding (NE/K002465/1 and NE/K002570/1) as part of the Greenhouse gAs Uk and Global Emissions (GAUGE) project. For technical and logistical support we would like to thank the team of Viridor on site. Thanks to Andrew Brunton and John Naylor from Ground-Gas Solutions for providing data from their survey of the landfill site. Many thanks to Thorsten Warneke and Hella van Asperen from the Institute of Environmental Physics (IUP), University of Bremen, for their advise on the deployment of the

10   FTIR. We also thank Peter Somkuti for processing the meteorological data. This research used the ALICE High Performance Computing Facility at the University of Leicester.





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
