# Peer review of "CH4 emission estimates from an active landfill site inferred from a combined approach of CFD modelling and in situ FTIR measurements"

_Atmospheric Measurement Techniques, 2016_

## Referee Comment (RC1) · W. Eugster (Referee) · 1 May 2017

The authors propose a method to model $CH_4$ emissions based on downwind concentration measurements from a landfill site. The main focus is on the application on a short field campaign of a few days.

**Major points**

**1.** The authors propose a method to model $CH_4$ emissions based on downwind concentration measurements from a landfill site. Since the paper focuses strongly on the

landfill site aspect, similar methods from other types of $CH_4$ sources are not included in the comparison. In principle each existing method could be considered a new method if applied to a different source than before, and all adaptations necessary to do so suggest it is a new method. However, for the reader it would be beneficial to get a better hierarchical overview over the general type of a method (independent of instruments used and specific tracers employed) and what is new/different/improved over existing methods. For example, there is a paper by Yver-Kwok et al. (2015, doi:10.5194/amt-8-2853-2015) that uses similar instrumentation but a different source (waste water treatment) but not in combination with modeling. And then there are methods strongly used for estimating $NH_3$ sources using downwind concentration measurements in a similar way, but maybe not specifically for $CH_4$ and using different instrumentation (e.g., Bell et al. 2016, doi:10.5194/amt-2016-350). It would thus really be desirable to get a broader overview over these methods and how the new proposed method differs from existing methods.

**2.** The use of the open source OpenFOAM software platform (I did not know this but it seems to be a good open source alternative to Comsomol) is interesting and thus making model code associated with this paper available to others would be a real benefit. This would in fact be the best option to increase reproducibility of the study. With the brief information about the model setup I would not be able to set up OpenFOAM in a way that corresponds to what the authors did.

**3.** Table 2: I do not really understand the percentage (with one decimal!) of the uncertainty: if a flux is 0.99 $\pm$ 0.39 and $\pm$ 0.39 denotes the standard deviation, then the 95% confidence interval is 1.96$\times$0.39 or 0.76, thus the uncertainty of the flux is 0.76/0.99 or 77% (not 44.4–44.9%). If I correctly understood your precentages are assuming a 40% uncertainty of the model and thus you somehow put 4.4–4.9% on top, but I cannot follow here.

**4.** The inclusion of a secondary source area without additional measurements rises the question whether the difference between CFD model and measurements is not

simply an artefact of the turbulence parametrisation in OpenFOAM. According to Fig. 5a the domain of the model is only 1.2×0.7 $km^2$ (approx.) and thus turbulent mixing (at least the large eddy mixing) is most likely pure parametrisation, not a model result. At least turbulence cannot equilibrate with the roughness of the topography in such a small domain. I think alternative explanations besides the hypothesized existance of a second source should be mentioned in the manuscript. It appears that Section 3.4 is rather speculative, and the comparison between model and measurements shown in Fig. 9 do not suggest that this secondary source solved the discrepancy between model and measurements.

**5.** Unfortunately the comparison between model and measurements is limited by the narrow wind direction sector available for the comparison. This strongly suggests that measuring concentration with a mobile setup to fully cover the plume (as e.g. in Herndon et al. 2005, doi:10.1039/b500411j) would have substantial benefits even in this application. (basically, I do not fully agree with your take-home message on page 23, lines 1–3).

**In general** the study is nicely carried out and the language of the text is of high quality, thus my critique really addresses more the aspect of novelty of the method (for a methods-centered journal, to be clear) in comparison to similar approaches that may not have been used explicitly for land fill sites yet. The empirical part quantifying the fluxes looks OK, although I was not quite clear whether I understood your approach to uncertainty estimations.

My recommendation: major revisions

**Details**

p2/l20: use minus sign in –0.00154
p2/l20: use USA for country specification

p3/l5: then $\rightarrow$ than “wider area than”
Fig. 3 (and elsewhere): use scientific/ISO8601 date and time notation (21:00 not 9 pm; 06:00 not 6 am); rather use the term “panels” for the two components of the “graph”
Table 1: “slope of the correlation”: a correlation has no slope, you mean “slope of the regression”
Fig. 4: “$CH_4$ distribution” is misleading, you show $\Delta CH_4$ – please adjust the wording.
Eq. (1): I find the multiplier ($10^6$ ppm) confusing. I think it is correct to leave that away and know that such a ratio is easier to report in percent, permil, ppm or whatever (this is not a unit conversion it is only a way how to express ratios)

---

## Referee Comment (RC2) · Anonymous Referee #2 · 13 May 2017

**GENERAL COMMENTS**

The authors propose a study to infer methane emissions from sub-areas of a landfill site by using a computational fluid dynamics model, and they present a short field campaign as a dataset for the model validation.

The paper is well written, the introduction addresses the background issue satisfactorily, but the paper objectives (not the project ones) could be better clarified; similarly, the novel character of the work presented does not get through within the paper.

My main concern with this paper is that the observations presented are very few: the data are generally well described, but I think they are not sufficient for a full characterisation and validation of the proposed model: for example, there is no data for describing meteorology-driven variations, and so on. However, this may not necessarily be the main focus of this paper, and besides, the presented work is of clear interest to the scientific community.

Measurements over three sampling days in the year can be considered a spot- measure, useful to verify rather than characterise an emission source: chemical reactions in the substrate and subsequent emissions can be driven by changing atmospheric pressure and temperature, for example, not only by the daily development of turbulence.

I see the presented work as mainly a modelling work: I think more emphasis should be put on the main advantages of the CFD models compared for example to backward Lagrangian models. I think this issue is touched upon in the abstract, but not in the conclusion, where it could be expanded. Also, the usage of LIDAR data and people surveying the site could be expanded in the method section (or in the conclusion). From what emerges from the results, the model seems fit for representing emissions in conditions of well-developed turbulence regime: however common this could be, it is a big limitation, and should be addressed in the conclusions, perhaps including criteria for good functioning of the model vs bad.

Overall, I think this work is well done and useful, but I recommend major revisions to be made.

SPECIFIC COMMENTS:

The description of the experimental site and of the FTIR and CFD methods, including the setup, is clear and well detailed; I think adding information on the dump age (the different sections of it) would be a benefit. The section with the background measurements would benefit from a better explanation on how the measurements were used, or explain it better in the results section.

P6 L6: wind is not a fluid, air is.

P7 L10-15. I agree with the authors that the emissions from the hot spot areas are not representative of the landfill site, however I believe omitting those measurements does introduce a bias as well, in that they will be taken into account as much lower emission areas. For a model validation they may not be suitable, but under an observational point of view they should not be ignored. Perhaps you could expand on this point.

P8 L10-13. I think that here it is not clear why you need an enhancement factor rather than an emission ratio: what are the advantages of the technique you're using? Adding explanations would help the reader in understanding the value of your work.

P8 L10-15. It is not clear here when you did use the background measurements and when not: is it only for CO2? Is it only for some calculations? Explain better.

P8 L20. "wind field" instead of "wind".

P8 L28-30. Can you really conclude this from your data? Perhaps change the wording highlighting this is a possible interpretation.

P10 Table 1. The slope is an outcome of a regression, not a correlation.

P11 L8. Substitute "emissions" with "emitted gases".

The section on "methane distribution" is not very conclusive: what is the message here?

P12 L9. Molar mass density, not mass concentration.

P13 L17. Delete "are" after fluxes.

P13 L20. Any suggestions on what these extra sources could be?

P13 L27. Why do you think night fluxes should be higher? How is the production (emission of methane) connected to day/night pattern?

P13 L31-32. It is good to show all data for completeness, but it would be very useful to

have possible explanations for peculiar data, or just further discussion.

P18 Figure8. Specify that concentration refer to the portakabin location as well.

P18 L14-18 This really would be sorted with a longer period of measurements. . .

P20 L5-10. This is not fully clear: are you suggesting that a larger area is wrong for there are e.g. roads in between, or non-emitting areas that are considered emitting? What would the suggestion be, if this is the case, and spell it out.

P20 L17-19. Specify the meteorological measurements are easy and can be maintained over long periods.

P20 L20. "stable" has a definite meaning when talking about atmospheric processes (I refer to stability parameters), and I am not sure you mean this here.

P21 L1-3. This last sentence seems to be there without having any evidence to support it.

---

## Author Response (AR1)

W. Eugster (Referee) werner.eugster@usys.ethz.ch Received and published: 1 May 2017

The authors propose a method to model CH4 emissions based on downwind concentration measurements from a landfill site. The main focus is on the application on a short field campaign of a few days.

**Major points**

**1.** The authors propose a method to model CH4 emissions based on downwind concentration measurements from a landfill site. Since the paper focuses strongly on the landfill site aspect, similar methods from other types of CH4 sources are not included in the comparison. In principle each existing method could be considered a new method if applied to a different source than before, and all adaptations necessary to do so suggest it is a new method. However, for the reader it would be beneficial to get a better hierarchical overview over the general type of a method (independent of instruments used and specific tracers employed) and what is new/different/improved over existing methods. For example, there is a paper by Yver-Kwok et al. (2015, doi:10.5194/amt-8-2853-2015) that uses similar instrumentation but a  $\Delta$ different source (waste water treatment) but not in combination with modeling. And then there are methods strongly used for estimating NH3 sources using downwind concentration (e.g., Bell et al. 2016, doi:10.5194/amt-2016-350). It would thus really be desirable to get a broader overview over these methods and how the new proposed method differs from existing methods.

**The introduction was extended to give a broader overview over similar approaches for emission estimates using dispersion models and also covering other gases of interest.**

Especially the use of dispersion modelling with respect to other target compounds and other sources (e.g. Bell et al. 2016) and the deployment of an FTIR in another set up (Yver-Kwok et al. 2015) has been mentioned. Additionally, another mobile sensor platform (UAS) has been included (Allen et al. 2014, 2016). It is emphasized that still a great effort is put into establishing the most appropriate sampling approach and measurement techniques for site-wide flux measurements of landfills.

**2**. The use of the open source OpenFOAM software platform (I did not know this but it seems to be a good open source alternative to Comsomol) is interesting and thus making model code associated with this paper available to others would be a real benefit. This would in fact be the best option to increase reproducibility of the study. With the brief information about the model setup I would not be able to set up OpenFOAM in a way that corresponds to what the authors did.

One of the main advantages of OpenFOAM is the open source availability, which makes it attractive to use. One of the downside is that it requires specific computational skill (linux based platform coded in C++).

Unfortunately, the authors were not in a position to release the code used in this study. However, we agree that the OpenFOAM description section was too limited to allow other users to reproduce this work.

**Therefore, a detailed section on model parameterisation was added in the manuscript (see section 2.4.2 Numerical settings).**

**3.** Table 2: I do not really understand the percentage (with one decimal!) of the uncertainty: if a flux is  $0.99 \pm 0.39$  and  $\pm 0.39$  denotes the standard deviation, then the 95% confidence interval is  $1.96 \times 0.39$  or 0.76, thus the uncertainty of the flux is 0.76/0.99 or 77% (not 44.4–44.9%). If I correctly understood your precentages are assuming a 40% uncertainty of the model and thus you somehow put 4.4–4.9% on top, but I cannot follow here.

We agree that the origin of the uncertainty in Table 2 was lacking explanation. The calculation of the uncertainty is based on the standard deviation of the averaged methane mole fraction of each bin (as shown in Fig. 6), not on the standard deviation of the calculated fluxes.

**The paragraph in section 3.3 (p13, ll13-16) was revised and the way that the uncertainty was calculated described more clearly. The caption of Table 2 was changed.**

**4.** The inclusion of a secondary source area without additional measurements rises the question whether the difference between CFD model and measurements is not simply an artefact of the turbulence parametrisation in OpenFOAM. According to Fig. 5a the domain of the model is only 1.2×0.7 km2 (approx.) and thus turbulent mixing (at least the large eddy mixing) is most likely pure parametrisation, not a model result. At least turbulence cannot equilibrate with the roughness of the topography in such a small domain. I think alternative explanations besides the hypothesized existance of a second source should be mentioned in the manuscript. It appears that Section 3.4 is rather speculative, and the comparison between model and measurements shown in Fig. 9 do not suggest that this secondary source solved the discrepancy between model and measurements.

Chamber measurements (results not shown) on the side area between the active site and the ridge detected additional, irregular methane emissions. This area was initially not taken into account, because we were focussing on the active site, but was considered as the secondary source area.

The turbulent parameterisation of OpenFOAM made use of a standard turbulent dispersion (Sct = 0.7, see section 2.4.2). This parameter can be changed in order to control the amount of turbulent mixing (lower Sct leads to greater turbulent mixing and vice-versa). Lowering the Sct would increase the width of the Gaussian shape in Fig. 5b, however the strength in concentrations would decrease as well. The authors were therefore keen to keep a standard dispersion setting as the CFD model was initially evaluated using very similar parameterisation (see Jeanjean et al. 2015; Jeanjean et al. 2017).

The boundary conditions were setup using the terrain roughness, hence the wind entering the computational domain is already accounting for roughness turbulent mixing. Another reason for the discrepancy between model and measurements would likely be the time averaged assumption used (here 3 minutes aggregated wind and concentrations data). During the aggregated time period, wind speeds and directions will fluctuate, emissions from the landfill are likely to oscillate as well, which could explain the difference found. This is addressed by using at least 5 3-min data points per bin for calculation of the fluxes.

A paragraph was added at the end of section 3.4 to address this point.

**5.** Unfortunately the comparison between model and measurements is limited by the narrow wind direction sector available for the comparison. This strongly suggests that measuring concentration with a mobile setup to fully cover the plume (as e.g. in Herndon et al. 2005, doi:10.1039/b500411j) would have substantial benefits even in this application. (basically, I do not fully agree with your take-home message on page 23, lines 1–3).

The referee is right pointing out, that having only one sampling point is a drawback in terms of sampling plume coverage. This is a limitation of this method as sampling is dependent on the right wind direction. A mobile application would allow for a broader sampling of the plume like it is done in tracer release experiments. A disadvantage there is the requirement of moving the sensor around, which does not allow for longer term/continuous sampling. For future applications one could make use of the multiple inlets of the FTIR by spreading sampling lines along the cross section of the plume or even at different heights and sampling in turns from there. Data from the CFD could be extracted to match the different sampling spots.

The CFD modelling is not a suitable tool to describe the concentrations on a high temporal resolution. For a good representation of the distribution of emissions it needs a few minutes of integration time.

Some text on plume coverage has been included in the summary and conclusions. P 23, lines 1-3 have been removed and the position of the sensor is mentioned further up in this section in the context of the definition of the source area.

**In general** the study is nicely carried out and the language of the text is of high quality, thus my critique really addresses more the aspect of novelty of the method (for a methods-centered journal, to be clear) in comparison to similar approaches that may not have been used explicitly for land fill sites yet. The empirical part quantifying the fluxes looks OK, although I was not quite clear whether I understood your approach to uncertainty estimations.

My recommendation: major revisions

**Details**

p2/l20: use minus sign in -0.00154 *Done.*p2/l20: use USA for country specification *Changed.*p3/l5: then ! than "wider area than" *Corrected.*Fig. 3 (and elsewhere): use scientific/ISO8601 date and time notation (21:00 not 9 pm; 06:00 not 6 am); rather use the term "panels" for the two components of the "graph" *Changed.*Table 1: "slope of the correlation": a correlation has no slope, you mean "slope of the regression" *Improved.*Fig. 4: "CH4 distribution" is misleading, you show ΔCH4 – please adjust the wording. *Changed in the figure caption and title of section.*Eq. (1): I find the multiplier (106 ppm) confusing. I think it is correct to leave that away and know that

such a ratio is easier to report in percent, permil, ppm or whatever (this is not a unit conversion it is only a way how to express ratios) *Removed.*  We would like to thank the referee for the valuable comments and their time to review the manuscript. While the referee comments were kept in black, the author comments are in blue. The italic font indicates where changes are made.

The authors propose a study to infer methane emissions from sub-areas of a landfill site by using a computational fluid dynamics model, and they present a short field campaign as a dataset for the model validation.

The paper is well written, the introduction addresses the background issue satisfactorily, but the paper objectives (not the project ones) could be better clarified; similarly, the novel character of the work presented does not get through within the paper.

My main concern with this paper is that the observations presented are very few: the data are generally well described, but I think they are not sufficient for a full characterisation and validation of the proposed model: for example, there is no data for describing meteorology-driven variations, and so on. However, this may not necessarily be the main focus of this paper, and besides, the presented work is of clear interest to the scientific community.

We agree that longer term measurements would need to be carried out to cover a wider range in meteorological conditions and to build up a larger data base for a full validation of the model. In the context of this field campaign, it was not possible to extend the measurement period. Therefore the focus was rather a feasibility study for the proposed method.

Measurements over three sampling days in the year can be considered a spot- measure, useful to verify rather than characterise an emission source: chemical reactions in the substrate and subsequent emissions can be driven by changing atmospheric pressure and temperature, for example, not only by the daily development of turbulence.

The study presented here rather has a focus on the method combining CFD with in situ measurements to derive fluxes, than to asses the whole landfill emissions from that site. The referee is right, that parameters like pressure and temperature can have an effect on landfill emissions. For that the measurements would need to be run longer or at different times of the year for short periods. But it was shown that this would potentially be possible with this approach.

**The summary was extended to include discussion of the measurement period.**

I see the presented work as mainly a modelling work: I think more emphasis should be put on the main advantages of the CFD models compared for example to backward Lagrangian models. I think this issue is touched upon in the abstract, but not in the conclusion, where it could be expanded.

To address Referee #2's concerns, paragraphs were added in the introduction and section 2.4 to discuss the differences/advantages of using CFD models against other dispersion models. A note was also added to the conclusions.

Also, the usage of LIDAR data and people surveying the site could be expanded in the method section (or in the conclusion).

**The description of the LIDAR data collection deserved to be more detailed, section 2.4.1 was added therefore in the manuscript and people acknowledged for carrying out the surveying work.**

From what emerges from the results, the model seems fit for representing emissions in conditions of well-developed turbulence regime: however common this could be, it is a big limitation, and should be addressed in the conclusions, perhaps including criteria for good functioning of the model vs bad.

That's correct, the model ideal conditions needs to be emphasized in the manuscript. The model best performs for wind speeds greater than 2 m s-1 and stable wind conditions. On the contrary, unsteady wind and low wind speeds are the worst conditions.

The authors decided to add a new section (2.4.3) to describe the model limitations.

Overall, I think this work is well done and useful, but I recommend major revisions to be made.

**SPECIFIC COMMENTS:**

The description of the experimental site and of the FTIR and CFD methods, including the setup, is clear and well detailed; I think adding information on the dump age (the different sections of it) would be a benefit. The section with the background measurements would benefit from a better explanation on how the measurements were used, or explain it better in the results section.

P6 L6: wind is not a fluid, air is.

**Changed to air.**

P7 L10-15. I agree with the authors that the emissions from the hot spot areas are not representative of the landfill site, however I believe omitting those measurements does introduce a bias as well, in that they will be taken into account as much lower emission areas. For a model validation they may not be suitable, but under an observational point of view they should not be ignored. Perhaps you could expand on this point.

P7 L15 – P8 L3: has been rephrased to point out that these hotspot emissions contribute to the total emissions of the landfill and are taken into account with the secondary source area (section 3.4), but that the measurements in close proximity to them were not suitable for a separate flux estimation approach of the active site with the CFD model.

P8 L10-13. I think that here it is not clear why you need an enhancement factor rather than an emission ratio: what are the advantages of the technique you're using? Adding explanations would help the reader in understanding the value of your work.

The enhancement factor should be equal to the emission ratio as long as there are no additional sources or dilution of the plume during transport to the FTIR. The term enhancement factor is used to emphasise that we did not measure directly at the source.

P8 L10-15. It is not clear here when you did use the background measurements and when not: is it only for CO2? Is it only for some calculations? Explain better.

Background values were not available for CO2 for the whole period. That's why we chose to determine the enhancement factor from the slope of the regression of CH4 to CO2. So, no background measurements were used for determination of the enhancement factor.

**P8, L10-16 were rephrased.**

P8 L20. "wind field" instead of "wind". Changed.

P8 L28-30. Can you really conclude this from your data? Perhaps change the wording highlighting this is a possible interpretation.

"These ratios are still representative of waste degradation under aerobic conditions, but show a higher CH4 content compared to the EF observed at the portakabin."

**Changed to**

"Compared to the EF observed at the portakabin they show a higher CH4 content, but can still be interpreted as being representative of waste degradation under mainly aerobic conditions."

P10 Table 1. The slope is an outcome of a regression, not a correlation.

**Corrected.**

P11 L8. Substitute "emissions" with "emitted gases". Changed.

**The section on "methane distribution" is not very conclusive: what is the message here?**

The section is supposed to give an overview of the methane enhancement (after subtracting the background) observed at the portakabin depending on the wind direction. The reader gets familiarised with this kind of representation of the data, which is further on used when the fluxes are calculated. Additionally, it shows that the observed enhanced methane comes only from the direction of the landfill site and that highest concentrations correspond to low wind speeds.

This section has been slightly reorganised to make these points more clear.

**P12 L9. Molar mass density, not mass concentration.**

In this context, either mass concentration or density can be used (https://doi.org/10.1351/goldbook.M03713). As it refers to the methane concentration, which is then converted to the mole fraction, we decided to change the symbol back to C. This is also used in the newly added Eq. (4), section 2.4.2.

P13 L17. Delete "are" after fluxes. Done.

**P13 L20. Any suggestions on what these extra sources could be?**

Hotspots along the side area between the slope and the active site have been observed, but were initially not taken into account, because we focussed on emissions from the active site. The sentenced was rephrased, "unknown source" was not the appropriate expression here.

**P13 L27. Why do you think night fluxes should be higher? How is the production (emission of methane) connected to day/night pattern?**

During daytime new waste is deposited on the active site and vehicles drive there and shift waste around. This mixes fresh air into the top layer and could lead to increased oxidation, while at night methane production could be favoured.

Decrease in temperatures over night could also result in higher methane emissions, when methane oxidising bacteria are less active. A small inverse temperature relationship was found by Riddick et al. 2016 for this landfill site.

Same CFD run used for day and night, which was optimised for daytime conditions. Change in atmospheric stability and turbulence could lead to artefacts in the results. From Antoine's experience the model results don't change much for night, unless the boundary layer is very low, e.g. in winter.

More data would be needed to investigate the effects on the nighttime emissions.

**A paragraph was added for discussion in section 3.3.**

P13 L31-32. It is good to show all data for completeness, but it would be very useful to have possible explanations for peculiar data, or just further discussion.

A paragraph was added to the manuscript in section 3.3.

P18 Figure8. Specify that concentration refer to the portakabin location as well.

Done.

P18 L14-18 This really would be sorted with a longer period of measurements...

**Yes. A comment was added at the end of the paragraph.**

P20 L5-10. This is not fully clear: are you suggesting that a larger area is wrong for there are e.g. roads in between, or non-emitting areas that are considered emitting?

**What would the suggestion be, if this is the case, and spell it out.**

Estimating the actual emitting area at the landfill is quite difficult as the terrain is very heterogeneous. Our focus was on the open active site as the main emission area, while Riddick et al. 2016 took a more generous approach by including the surrounding area. As is described in the manuscript, the surrounding area also contributes to the overall emission, but has a lower emission strength.

**An explanation was added to section 3.5.**

P20 L17-19. Specify the meteorological measurements are easy and can be maintained over long periods.

**Done.**

P20 L20. "stable" has a definite meaning when talking about atmospheric processes (I refer to stability parameters), and I am not sure you mean this here.

*Changed to: "Consistent fluxes from the active site were found for three different days with southerly winds transporting air from the source area towards the portakabin"*

**P21 L1-3. This last sentence seems to be there without having any evidence to support it.**

These lines have been removed. The position of the instrument is now discussed a bit higher up in the summary.

**CH4 emission estimates from an active landfill site inferred from a combined approach of CFD modelling and in situ FTIR measurements**

Hannah Sonderfeld1, Hartmut Bösch1,2, Antoine P.R. Jeanjean1, Stuart N. Riddick3, Grant Allen4, Sébastien Ars5, Stewart Davies6, Neil Harris7, Neil Humpage1, Roland Leigh1, and Joseph Pitt4 1Earth Observation Science Group, Department of Physics and Astronomy, University of Leicester, Leicester, UK. 2National Centre for Earth Observation, University of Leicester, Leicester, UK 3Department of Civil and Environmental Engineering, Princeton University, NJ, United States 4Centre for Atmospheric Science, The University of Manchester, Manchester, UK. 5Laboratoire des Sciences du Climat et de l'Environnement (LSCE/IPSL), CNRS-CEA-UVSQ, Université de Paris-Saclay, Gif-sur-Yvette, France 6Viridor Waste Management Limited, Peninsula House, Rydon Lane, Exeter, Devon, UK 7Centre for Atmospheric Informatics and Emissions Technology, Cranfield University, Cranfield, UK *Correspondence to:* Hannah Sonderfeld (hs287@le.ac.uk)

Abstract.

[revised manuscript text omitted]

- 10 speed (Lohila et al., 2007). They are best suited for flat terrain and have difficulties with complex topography. Sensors on mobile platforms offer the advantage of a wider coverage of the emission plume and a more flexible sampling strategy which can be adapted depending on the wind direction. In recent years tracer dispersion methods were developed and became more widely used (Czepiel et al., 1996; Galle et al., 2001; Foster-Wittig et al., 2015; Mønster et al., 2015). In this approach a tracer is released at the source and sampled downwind together with the target gas. Initially, sulfur hexafluoride
- 15  $(SF_6)$  (Czepiel et al., 1996) and N2O (Galle et al., 2001) were used as tracer, which are greenhouse gases themselves. Mønster et al. (2014) and Foster-Wittig et al. (2015) used acetylene as a tracer, which was co-measured with CH4 with cavity ring-down spectroscopy (CRDS). This technique provides accurate measurements of CH4 emissions of landfills and can also be applied to divide between several sources in one area by using an additional tracer (Scheutz et al., 2011; Mønster et al., 2014). A requirement for this method is accessibility downwind of the site for sampling the plume and the time span that can be covered
- 20 is limited. The use of an unmanned aerial system (UAS) as a mobile sampling platform has been carefully assessed recently (Allen et al., 2014, 2016). Present challenges are to find high precision  $CH_4$  sensors that can be installed and operated on an UAS and to develop a safe flight pattern covering the up- and downwind signal (Allen et al., 2016).

Atmospheric dispersion models appear as a useful tool for investigation of landfill site emissions from landfills and other area sources. Delkash et al. (2016) used a forward model to analyse the effects of wind on short term variations in

- 25 landfill emissions in combination with a tracer method. Previously, Hrad et al. (2014) applied an inverse dispersion technique to The use of backward Lagrangian modelling for estimating gaseous emissions from a known area source in flat terrain with a single sensor has been described in detail by Flesch et al. (1995, 2004). This technique was also applied by Bell et al. (2017) for monitoring ammonia emissions from grazing cattle. Hrad et al. (2014) used backward Lagrangian modelling to estimate emissions from an open windrow composting plant. They found an agreement of 10 to 30 % in an inter-comparison to tracer release
- 30 experiments over five days. Zhu et al. (2013) and Riddick et al. (2016) applied this method for monitoring  $CH_4$  emissions from a landfill site.

The GAUGE (Greenhouse gAs Uk and Global Emissions) project aims for a better understanding and quantification of the UK GHG budget to support GHG emission reduction measures. In this context a two week field campaign between 4 and 15 August 2014 at a landfill site north of Ipswich, UK, was conducted as part of the GAUGE project to improve our

35 understanding of landfill emissions and to investigate different methods for flux quantification. Here, we present simultaneous

and continuous observation of  $CO_2$  and  $CH_4$  with in situ Fourier Transform Infrared (FTIR) spectroscopy at this landfill site. The use of the same kind of instrument for measurements of emissions from a waste water treatment plant was presented by Yver Kwok et al. (2015) in combination with floating chambers on the basins.

The application of a Computational Fluid Dynamics (CFD) model to the point measurements for estimating  $CH_4$  fluxes

- 5 is described and assessed. For complex terrains like a landfill site CFD models are expected to be more useful compared to Gaussian tools (Mazzoldi et al., 2008). Topographic information can be used by the CFD model to adapt to a more complex terrain, where backward Lagrangian models work best on a horizontally homogeneous surface layer (Flesch 
[revised manuscript text omitted]

- 25 Through resolving three-dimensional distributions of wind flow and gas concentration they provide space filling results, which makes them an attractive choice compared to Lagrangian models. The CFD simulations presented in this study have been validated previously by a comparison exercise against a wind tunnel experiment (Jeanjean et al., 2015) and measurements from an urban monitoring station (Jeanjean et al., 2017). As a result of this comparison it was shown that a model accuracy of 30 % to 40 % can be achieved. This represents a slight amelioration in respect to traditional Gaussian dispersion modelling. The CFD
- 30 simulations were performed under the OpenFOAM software platform. For calculating the wind flow, the Reynolds-averaged Navier-Stokes (RANS)k  $\epsilon$  model (Launder et al., 1975) was used. The dispersion of emissions from

**2.4.1 Landfill site survey and computational domain**

This study made use of a digital surface model, which was obtained from a terrestrial LIDAR (Light Detection and Ranging) survey, collected using a terrestrial laser scanner (Riegl LMSZ420i). The data was collected with a point spacing of between 20 and 50 cm depending on the accessibility of the landfill sitewas simulated with a passive scalar transport equation (for full flow and boundary conditions see Jeanjean et al. (2015)). LIDAR scans from five locations around the site were then merged into a single surface model element using the Innovmetrics PolyWorks software. The landfill surface data was finally

5 geo-referenced with a differential GPS (Global Positioning System, Trimble Pro 6T) which provides a submeter accuracy for global georeferencing. A more detailed summary of the use and processing of this kind of LIDAR data can be found in Hodgetts (2013).

A wall function was used to define the boundary conditions for the ground reproducing the landfill surface roughness. The landfill terrain was modelled with a roughness length value of 0.03 m, which corresponds to an open terrain with grass and

10 a few isolated obstacles (WMO, 2008). A The resulting digital surface model was then resampled into a 1 m grid, which in turn was extended using a 5.0 m digital elevation model from the Ordnance Survey (UK government agency responsible for topographic survey and mapping of Great Britain) to extend the studied area as shown in Fig 5 (a). The terrain was then incorporated as a 3 dimensional file to build a computational grid in the OpenFOAM CFD software.

The total number of cells used for the simulation numbered 142 000 cells was used for this simulation. Boundaries 000.
15 The boundaries used for the mesh were set to are (in British National Grid, minimum to maximum): X=[610350,-611650], Y=[249700,-250500], Z=[0,-500]with. The initial cells of the domain were assigned a dimension of 30 m. The cells corresponding to the terrain (ground) were assigned a size of 2 m and were kept constant up to 30 m away from the ground. Their resolution was then coarsened beyond 30 m with a maximum expansion ratio of 1.2. Topographic information for-

**2.4.2 Numerical settings**

20 The wind flow in the CFD model were gained from a LIDAR (Light Detection And Ranging) survey was calulated with the Reynolds-averaged Navier-Stokes (RANS) k -  $\epsilon$  model (Launder et al., 1975). Following a parametrisation for a neutral atmospheric boundary layer in Hargreaves and Wright (2007), the mean velocity boundary flow and the turbulent dissipation were set up to follow a logarithmic law using the ABLInletVelocity U (Eq. 1) and ABLInletEpsilon  $\epsilon$  (Eq. 2) utilities in OpenFOAM such that

$$\quad U = \frac{U^*}{K} ln\left(\frac{z+z_0}{z_0}\right)$$

and

$$\epsilon = \frac{U^{*3}}{Kz} \left(1 - \frac{z}{\delta}\right),$$

(1)

(2)

where K is the Karman's constant, z is the height coordinate (m),  $z_0$  is the roughness length (m),  $\delta$  is the boundary layer depth (m) and U\* is the frictional velocity (m s-1). The turbulent kinetic energy k was setup as follows

$$k = \frac{U^{*2}}{\sqrt{C_{\mu}}},\tag{3}$$

5

where  $C_{\mu} = 0.09$  is a k- $\epsilon$  constant.

The top boundary condition of the domain was setup as a symmetry condition. The inlets, where air enters the domain, and outlets, where air leaves the domain, were adjusted depending on the simulated wind conditions. For example, to simulate a southeasterly wind, the two inlets would be the south and eastern sides of the landfill site. At its borders the LIDAR map was extended with a 5 m digital elevation model (Ordnance Survey) domain and the outlets would be the northern and western

10 sides. A wall function was used for the ground to reproduce the landfill surface roughness. A roughness length value of 0.03 m was used to model the landfill terrain. This roughness length value corresponds to an open terrain with grass and a few isolated obstacles (WMO, 2008).

The dispersion of emissions from the landfill site was simulated using a passive scalar transport equation defined such that

$$\frac{\partial C}{\partial t} + \nabla (UC) = \nabla^2 \left( (D + K_e)C \right),\tag{4}$$

15 where C is the transported scalar (here  $CH_4$ , g m-3), U is the fluid velocity (m s-1), D is the diffusion coefficient (m2 s-1) and  $K_e$  is the eddy diffusion coefficient (m2 s-1). The eddy diffusion coefficient can be expressed as  $K_e = \mu_t/Sc_t$ , where  $\mu_t$ is the eddy viscosity or turbulent viscosity (m2 s-1) and Sct is the turbulent Schmidt number. The turbulent Schmidt number (Sct) values range between 0.3 to 1.3 (Tominaga and Stathopoulos, 2007), a Sct relatively common value of 0.7 was used.

**2.4.3 Model limitations**

- 20 A RANS CFD model provides a steady state view of the reality, which corresponds to a fixed picture of the wind flow and pollutant concentrations. In real life, the wind is oscillating in strength and directions and  $CH_4$  concentrations are highly variable following wind and landfill emission patterns. This study accounts for a calculated 3 minutes averaged concentration of  $CH_4$  and the use of this estimation introduces limitations in terms of temporal variation. The model used here was best suited for constant wind directions, RANS CFD model should be used with care when wind conditions are variable.
- Thermal effects can affect gas dispersion as well, especially for large temperature gradients and low wind speeds. For wind speeds greater than  $2 \text{ m s}^{-1}$ , previous studies have noted that wind dynamics are predominant over thermal effects which can then be neglected (Parra et al., 2010; Santiago et al., 2017). In this study, wind speeds greater than  $2 \text{ m s}^{-1}$  were used which justifies the assumption taken of an isothermal flow.

Despite these limitations, CFD dispersion models are currently one of the most advanced tools available for researchers to

30 model gas dispersion over non-uniform terrain. They are most suited for well-developed turbulence regime when stable wind directions and wind speeds conditions are met.

**3 Results and discussion**

The landfill campaign took place between 4 and 15 August 2014. Initially, wind was coming from northeast with relatively low wind speeds (see Fig. 2, top panel). On 8 and 10 August wind came mainly from east to southeast, while the dominant wind

- 5 direction on 9 and 11 to 12 August was from the south. At the end of the campaign the wind shifted more towards a westerly wind. The most frequent wind direction was around  $210^{\circ}$  ( $0^{\circ}/360^{\circ}$  corresponding to North) and wind speeds ranged from 0.1 to 13 m s-1. The time series of measured CH4 and CO2 mole fractions are shown in Fig. 2 in the lower two panels colour coded with the wind direction. The active site lies roughly between 170° and 240° as seen from the portakabin. CH4 values drop to background levels during measurements for air from the northern semi-circle (black and grey lines in Fig. 2), in the
- 10  $CO_2$  data a constantly low background value does not become apparent. High peaks in both gases appear before midnight on 8 August, when wind speeds were dropping to near zero, and in the following night for wind directions of 150° to 190°, which is only partially influenced by the active site. Two periods with wind constantly coming from the active area occurred during the course of the campaign: 9 August and 11 to 12 August. Air influenced by the active site was also measured during the night of 9 to 10 August until after midnight and on 14 August from the early morning hours to noon. These periods were less stable in
- 15 wind direction compared to the former time periods.

---

## Editor Decision (ED1)

**Editor Comments to the revised version of amt-2016-382**

In the following list, the page (P) and line (L) numbers generally refer to the revised version with track changes. But it has to be noted that the line numbers from page 11 on were somehow confounded and thus I used the true line number per page.

**Scientific Comments**

1) In the response to Referee W. Eugster, the authors write that they "were therefore keen to keep a standard dispersion setting as the CFD model was initially evaluated using very similar parameterisation…".
However, on P17, L14-15 it is stated that the turbulence mixing parameters of the model were optimized to match the bag samples of a tracer release experiment at the landfill site. This are two contradicting statements in my view that need explanation. In addition, the optimization of the turbulence parameterization and the performed tracer release experiment need to be described in more detail because they are quite crucial for the presented results.

2) P6, L17-18: Indicate the height of the wind measurements. How accurate was the wind direction obtained with the instrument/setup?

3) P8, L25-28: I do not agree with the conclusion here, that thermal effects can be generally neglected for wind speeds greater than 2 m s$^{-1}$. The two cited literature references do not provide enough arguments for the conclusion. Both references report on studies in winter, when thermal effect (especially unstable situations) do not have the same importance like for the present summer time experiment. Also the urban source distribution and environmental conditions were different from the present study. Moreover, the height of the wind measurement (also in relation to the underlying surface roughness) needs to be taken into account. Therefore the text needs to be rephrased here and the (non-negligible) uncertainty of the CFD model related to unaccounted thermal stability effects should be mentioned (or better arguments need to be provided for the original conclusion).

4) P26, L30-32: It may be useful to mention here, that the use of multiple, spatially distributed sampling points would allow to better identify and distinguish between different source areas, (more accurately than via wind direction variations).

5) P26, last line: Please specify how emissions can be detected by an "initial walk over survey".

**Technical and Language Corrections**

- P1, L11: specify: "…corresponding to a spatially integrated emission of 53.3 kg h$^{-1}$ …"

- P5, L1: In the title of Section 2.2 better use "FTIR" instead of "Spectronus". It would be more informative, because "FTIR" is used throughout the text.

- P6, L25: The formulation "…provide space filling results, …" is unclear. Rephrase this sentence. Lagrangian models also provide a spatially resolved 3D distribution of the concentration plume but usually cannot account for topography effects on the wind field. This should be clarified here.

- P6, L26: The sentence "The CFD simulations presented in this study have been validated previously by …" is unclear to me. I assume that the CFD model in general has been validated, not the specific CFD simulations in this study. Please specify and rephrase.

- P8, L15: correct to "where C is the concentration of …"

- P8, L15: "D" should be better specified e.g. as "molecular diffusion coefficient".

- P8, L18: The last sentence of section 2.4.2 should be shortened to "$Sc_t$ values range between…".

- Figure 3 caption: the expression "…as the gradient from the correlation of …" is not adequate. Rephrase e.g. to "…as the linear regression slope of $\chi CH_4$ vs $\chi CO_2$ …" like in the main text.

- Figure 5a: Indicate, which coordinate system was used on the axes?

- P23, L1: omit "derived"

- P26, L1: correct to "They report an uncertainty of 42% that is similar to our approach."

- P27, L2: rephrase the sentence "The main uncertainty results from the model accuracy."

- P27, L4: The formulation "…from a range in wind direction" is not clear to me. Please rephrase.

- P27, L11: change to "Enhanced $\Delta CH4$ was observed …"

---

## Author Response (AR2)

We would like to thank the referee and the editor for their time to review the revised manuscript in combination with the author's response to the reviews. The specific comments helped to improve the manuscript considerably. While the referee and editor comments were kept in black, the author comments are in blue. Changes made to the manuscript are printed in italics.

**Editor Comments to the revised version of amt-2016-382**

In the following list, the page (P) and line (L) numbers generally refer to the revised version with track changes. But it has to be noted that the line numbers from page 11 on were somehow confounded and thus I used the true line number per page.

**Scientific Comments**

1)      In the response to Referee W. Eugster, the authors write that they "were therefore keen to keep a standard dispersion setting as the CFD model was initially evaluated using very similar parameterisation…".
However, on P17, L14-15 it is stated that the turbulence mixing parameters of the model were optimized to match the bag samples of a tracer release experiment at the landfill site. This are two contradicting statements in my view that need explanation. In addition, the optimization of the turbulence parameterization and the performed tracer release experiment need to be described in more detail because they are quite crucial for the presented results.

We agree with the editor that this statement was lacking clarity. A more accurate phrasing with regard to point 4) of the referee would have been: "The authors were keen on using a dispersion setting close to the value that was used in previous evaluation experiments (see Jeanjean et al. 2015; Jeanjean et al. 2017) and within the standard range of 0.3 – 1.3. An on-site tracer release experiment was in favour of using Sct = 0.7 (see Jeanjean 2017), which is close to the value of 0.5 used in the street canyon and wind tunnel experiments in Jeanjean et al. 2015; Jeanjean et al. 2017 and has also been used by Riddle et al. 2004 for CFD simulations over agricultural land.

*More details of the tracer release experiment are added to section 2.4.2 of the manuscript.*

*P17, L14-15 was shortened to "It should be noted that the CFD model turbulence mixing parameters were optimised to match the bag samples of the tracer release experiment."*

2)      P6, L17-18: Indicate the height of the wind measurements. How accurate was the wind direction obtained with the instrument/setup?

 - The wind measurements were taken 2 m above ground with a WindMaster Pro by Gill Instruments. The accuracies given by the manufacturer are 1 % RMS for the wind speed and 0.5 degree for the wind direction. *This information has been added to section 2.3.*

3)      P8, L25-28: I do not agree with the conclusion here, that thermal effects can be generally neglected for wind speeds greater than 2 m s$^{-1}$. The two cited literature references do not provide enough arguments for the conclusion. Both references report on studies in winter, when thermal effect (especially unstable situations) do not have the same importance like for the present summer time experiment. Also the urban source distribution and environmental conditions were different from the present study. Moreover, the height of the wind measurement (also in relation to the underlying surface roughness) needs to be taken into account. Therefore the text needs to be rephrased here and the (non-negligible) uncertainty of the CFD model related to unaccounted thermal stability effects should be mentioned (or better arguments need to be provided for the original conclusion).

After a deep literature review, no studies were found quantifying the effects of thermally induced turbulence in a rural environment or at a landfill site on CFD simulations. Almost all references found were investigating thermal effects in urban areas, especially looking at urban heat islands effects. No thermal mapping was carried out in this study, as it was originally thought to be of secondary importance. Nevertheless, the effect should decrease with increasing wind speed and therefore be limited.

*The text has been adapted to avoid misleading arguments, especially by mentioning that the previous references were for urban environment in winter.*

4)      P26, L30-32: It may be useful to mention here, that the use of multiple, spatially distributed sampling points would allow to better identify and distinguish between different source areas, (more accurately than via wind direction variations).

– A sentence was added to the conclusions: "*The presented method could be improved by using multiple, spatially distributed sampling points.*"

5)      P26, last line: Please specify how emissions can be detected by an "initial walk over survey".

– An initial walk over survey with a handheld sensor could be used to assess the extension of source areas and location of hotspots by mapping the methane concentration. *The sentence in the manuscript was changed to:" Chamber measurements or an initial walk over survey with a small portable CH4 sensor are valuable tools to characterise different parts of a landfill site and detect emission hotspots which are otherwise easily overseen by point measurements.*

**Technical and Language Corrections**

- P1, L11: specify: "…corresponding to a spatially integrated emission of 53.3 kg h$^{-1}$ …"
  Changed.

- P5, L1: In the title of Section 2.2 better use "FTIR" instead of "Spectronus". It would be more informative, because "FTIR" is used throughout the text. - Changed.

-       P6, L25: The formulation "…provide space filling results, …" is unclear. Rephrase this sentence. Lagrangian models also provide a spatially resolved 3D distribution of the concentration plume but usually cannot account for topography effects on the wind field. This should be clarified here. – The sentence has be rephrased to emphasise that CFD model are superior on dealing with complex wind fields on small scales compared to Eulerian and Lagrangian dispersion models: *"Resolving three-dimensional distributions of wind flow and gas concentration in the modelling domain on small scales makes them an attractive choice compared to Eulerian and Lagrangian dispersion models (Leelossy et al. 2014)."*

- P6, L26:  The sentence "The CFD simulations presented in this study have been validated previously by …" is unclear to me. I assume that the CFD model in general has been validated, not the specific CFD simulations in this study. Please specify and rephrase.  – The wording was not clear indeed. The CFD model has been evaluated. *The sentence was changed to: " The CFD model presented in this study … has previously been evaluated …"*

- P8, L15: correct to "where C is the concentration of …" - Changed.

- P8, L15: "D" should be better specified e.g. as "molecular diffusion coefficient". - Changed.

- P8, L18: The last sentence of section 2.4.2 should be shortened to "$Sc_t$ values range between…". - Changed.

- Figure 3 caption: the expression "…as the gradient from the correlation of …" is not adequate. Rephrase e.g. to "…as the linear regression slope of $\chi CH_4$ vs $\chi CO_2$ …" like in the main text. - Changed. *In the text χCH4 to χCO2 was changed to χCH4 versus χCO2.*

- Figure 5a: Indicate, which coordinate system was used on the axes? - The used coordinate system is the British National Grid. *This information was added to the caption.*

- P23, L1: omit "derived" - Done.

- P26, L1: correct to "They report an uncertainty of 42% that is similar to our approach." Done.

- P27, L2: rephrase the sentence "The main uncertainty results from the model accuracy." – The sentence has been *rephrased to "The main contribution to the uncertainty of the derived emissions results from the limitations of the CFD model simulations."* and moved further down in the conclusions.

- P27, L4: The formulation "…from a range in wind direction" is not clear to me. Please rephrase. – This has been rephrased.

- P27, L11: change to "Enhanced ΔCH4 was observed …" - Changed.

**Comments on the revised manuscript "CH$_4$ emission estimates from an active landfill site inferred from a combined approach of CFD modelling and in situ FTIR measurements" by Hannah Sonderfeld et al.**

**Anonymous Referee #2**
Submitted on 07 August 2017

The authors have addressed satisfactorily my comments, and therefore I recommend this second version of the manuscript for publication on AMT.

I think adding a reference or just an URL for the OpenFOAM model would be fair for acknowledgement, and I would insert it in section 2.4. – The url http://www.openfoam.com was added to the manuscript.

It is von Karman constant, correct its name (section 2.4.2) – Corrected.

[revised manuscript text omitted]

Atmospheric dispersion models appear as a useful tool for investigation of emissions from landfills and other area sources. Delkash et al. (2016) used a forward model to analyse the effects of wind on short term variations in landfill emissions in combination with a tracer method. The use of backward Lagrangian modelling for estimating gaseous emissions from a known area source in flat terrain with a single sensor has been described in detail by Flesch et al. (1995, 2004). This technique was also applied by Bell et al. (2017) for monitoring ammonia emissions from grazing cattle. Hrad et al. (2014) used backward Lagrangian modelling to estimate emissions from an open windrow composting plant. They found an agreement of 10 to 30 % in an inter-comparison to tracer release experiments over five days. Zhu et al. (2013) and Riddick et al. (2016) applied this method for monitoring $CH_4$ emissions from a landfill site.

The GAUGE (Greenhouse gAs Uk and Global Emissions) project aims for a better understanding and quantification of the UK GHG budget to support GHG emission reduction measures. In this context a two week field campaign between 4 and 15 August 2014 at a landfill site north of Ipswich, UK, was conducted as part of the GAUGE project to improve our understanding of landfill emissions and to investigate different methods for flux quantification. Here, we present simultaneous and continuous observation of $CO_2$ and $CH_4$ with in situ Fourier Transform Infrared (FTIR) spectroscopy at this landfill site. The use of the

same kind of instrument for measurements of emissions from a waste water treatment plant was presented by Yver Kwok et al. (2015) in combination with floating chambers on the basins.

[revised manuscript text omitted]

UK) throughout the campaign. The accuracy for the wind speed is 1 % RMS (root-mean-square) at 12 m/s and 0.5° in wind direction for typical wind speeds.

**2.4   CFD model**

The gas dispersion from the landfill surface was calculated with a CFD model using the OpenFOAM (Open Field Operation and Manipulation) open source software platform (freely available at http://www.openfoam.com). CFD models use fluid dynamics equations constrained by boundary conditions that are solved numerically to calculate the behaviour of a fluid such as air within a particular domain (here the landfill terrain). CFD models require a complex parametrisation compared to traditional Gaussian dispersion models, but they have been shown to provide increased accuracy over complex terrain (Buccolieri and Sabatino, 2011), which can be considered to be the case over the landfill site. Resolving three-dimensional distributions of wind flow and gas concentration in the modelling domain on small scales makes them an attractive choice compared to Eulerian and Lagrangian dispersion models (Leelőssy et al., 2014). The CFD model  presented in this study has previously been evaluated  by a comparison exercise against a wind tunnel experiment (Jeanjean et al., 2015) and measurements from an urban monitoring station (Jeanjean et al., 2017). As a result of this comparison it was shown that a model accuracy of 30 % to 40 % can be achieved. This represents a slight amelioration in respect to traditional Gaussian dispersion modelling.

**2.4.1   Landfill site survey and computational domain**

This study made use of a digital surface model, which was obtained from a terrestrial LIDAR (Light Detection and Ranging) survey, collected using a terrestrial laser scanner (Riegl LMSZ420i). The data was collected with a point spacing of between 20 and 50 cm depending on the accessibility of the landfill site. LIDAR scans from five locations around the site were then merged into a single surface model element using the Innovmetrics PolyWorks software. The landfill surface data was finally geo-referenced with a differential GPS (Global Positioning System, Trimble Pro 6T) which provides a submeter accuracy for global georeferencing. A more detailed summary of the use and processing of this kind of LIDAR data can be found in Hodgetts (2013).

The resulting digital surface model was then resampled into a 1 m grid, which in turn was extended using a 5.0 m digital elevation model from the Ordnance Survey (UK government agency responsible for topographic survey and mapping of Great Britain) to extend the studied area as shown in Fig 5 (a). The terrain was then incorporated as a 3 dimensional file to build a computational grid in the OpenFOAM CFD software.

The total number of cells used for the simulation numbered 142 000. The boundaries used for the mesh are (in British National Grid, minimum to maximum): X=[610350 611650], Y=[249700 250500], Z=[0 500]. The initial cells of the domain were assigned a dimension of 30 m. The cells corresponding to the terrain (ground) were assigned a size of 2 m and were kept constant up to 30 m away from the ground. Their resolution was then coarsened beyond 30 m with a maximum expansion ratio of 1.2.

**2.4.2  Numerical settings**

The wind flow in the CFD model was caluated with the Reynolds-averaged Navier-Stokes (RANS) k - $\epsilon$ model (Launder et al., 1975). Following a parametrisation for a neutral atmospheric boundary layer in Hargreaves and Wright (2007), the mean velocity boundary flow and the turbulent dissipation were set up to follow a logarithmic law using the ABLInletVelocity U (Eq. 1) and ABLInletEpsilon $\epsilon$ (Eq. 2) utilities in OpenFOAM such that

$$U = \frac{U^*}{K} ln \left( \frac{z + z_0}{z_0} \right) \tag{1}$$

and

$$\epsilon = \frac{U^{*3}}{Kz} \left( 1 - \frac{z}{\delta} \right), \tag{2}$$

where K is the von Karman's constant, z is the height coordinate (m), $z_0$ is the roughness length (m), $\delta$ is the boundary layer depth (m) and $U^*$ is the frictional velocity (m s$^{-1}$). The turbulent kinetic energy k was setup as follows

$$k = \frac{U^{*2}}{\sqrt{C_\mu}}, \tag{3}$$

where $C_\mu$ = 0.09 is a k-$\epsilon$ constant.

The top boundary condition of the domain was setup as a symmetry condition. The inlets, where air enters the domain, and outlets, where air leaves the domain, were adjusted depending on the simulated wind conditions. For example, to simulate a southeasterly wind, the two inlets would be the south and eastern sides of the landfill domain and the outlets would be the northern and western sides. A wall function was used for the ground to reproduce the landfill surface roughness. A roughness length value of 0.03 m was used to model the landfill terrain. This roughness length value corresponds to an open terrain with grass and a few isolated obstacles (WMO, 2008).

The dispersion of emissions from the landfill site was simulated using a passive scalar transport equation defined such that

$$\frac{\partial C}{\partial t} + \nabla(UC) = \nabla^2 \left( (D + K_e)C \right), \tag{4}$$

where C is the concentration of  CH$_4$ (g m$^{-3}$), U is the fluid velocity (m s$^{-1}$), D is the molecular diffusion coefficient (m$^2$ s$^{-1}$) and $K_e$ is the eddy diffusion coefficient (m$^2$ s$^{-1}$). The eddy diffusion coefficient can be expressed as $K_e = \mu_t / Sc_t$, where $\mu_t$ is the eddy viscosity or turbulent viscosity (m$^2$ s$^{-1}$) and Sc$_t$ is the turbulent Schmidt number.  Sc$_t$ values range between 0.3 to 1.3 (Tominaga and Stathopoulos, 2007). A suitable Sc$_t$ for this study was determined in a tracer release experiment on-site conducted by the University of Bristol. For details see Jeanjean (2017). Perfluoromethylcyclohexane (PMCH) was released from a point source

on the southern edge of the landfill site. While the wind was coming from a southern direction four bags were sampled on the northern part of the landfill. A relatively common value of $Sc_t = 0.7$ appeared to be the best choice to represent the measured concentrations of the bag samples. Riddle et al. (2004) also used $Sc_t = 0.7$ for CFD simulations over agricultural land.

**2.4.3 Model limitations**

A RANS CFD model provides a steady state view of the reality, which corresponds to a fixed picture of the wind flow and pollutant concentrations. In real life, the wind is oscillating in strength and directions and $CH_4$ concentrations are highly variable following wind and landfill emission patterns. This study accounts for a calculated 3 minutes averaged concentration of $CH_4$ and the use of this estimation introduces limitations in terms of temporal variation. The model used here was best suited for constant wind directions, RANS CFD model should be used with care when wind conditions are variable.

Thermal effects can affect gas dispersion as well, especially for large temperature gradients and low wind speeds. For wind speeds greater than $2 \, \mathrm{m \, s^{-1}}$, previous studies in an urban environment in winter have noted that wind dynamics are predominant over thermal effects which can then be neglected (Parra et al., 2010; Santiago et al., 2017). The authors are not aware of any studies which quantify thermal effects on CFD modelling in rural environments or on a landfill site. In this study, thermal effects were not taken into account in the CFD model and remain a source of uncertainty. But, since only wind speeds greater than $2 \, \mathrm{m \, s^{-1}}$ were used , the influence of thermal effects should be minimised.

Despite these limitations, CFD dispersion models are currently one of the most advanced tools available for researchers to model gas dispersion over non-uniform terrain. They are most suited for well-developed turbulence regime when stable wind directions and wind speeds conditions are met.

**3 Results and discussion**

The landfill campaign took place between 4 and 15 August 2014. Initially, wind was coming from northeast with relatively low wind speeds (see Fig. 2, top panel). On 8 and 10 August wind came mainly from east to southeast, while the dominant wind direction on 9 and 11 to 12 August was from the south. At the end of the campaign the wind shifted more towards a westerly wind. The most frequent wind direction was around 210° (0°/360° corresponding to North) and wind speeds ranged from 0.1 to $13 \, \mathrm{m \, s^{-1}}$. The time series of measured $CH_4$ and $CO_2$ mole fractions are shown in Fig. 2 in the lower two panels colour coded with the wind direction. The active site lies roughly between 170° and 240° as seen from the portakabin. $CH_4$ values drop to background levels during measurements for air from the northern semi-circle (black and grey lines in Fig. 2), in the $CO_2$ data a constantly low background value does not become apparent. High peaks in both gases appear before midnight on 8 August, when wind speeds were dropping to near zero, and in the following night for wind directions of 150° to 190°, which is only partially influenced by the active site. Two periods with wind constantly coming from the active area occurred during the course of the campaign: 9 August and 11 to 12 August. Air influenced by the active site was also measured during the night of

9 to 10 August until after midnight and on 14 August from the early morning hours to noon. These periods were less stable in wind direction compared to the former time periods.

Much higher mole fractions with up to 700 ppm $CO_2$ and over 100 ppm $CH_4$ were observed by the UGGA at the ridge. These particularly high values were measured before the FTIR measurements were started, so a direct comparison here is not possible. Towards the end of the campaign both instruments were operated at the same time. Mole fractions measured then were much lower compared to the beginning, but values at the ridge were still enhanced compared to the portakabin. Chamber measurements along the south side of the ridge leading down to the active site showed that the cover of the old landfill part was not leak tight and allowed for additional significant emissions. $CH_4$ migrating underneath the landfill cap can leak out at places where the landfill cover is interrupted, e.g. at the edge of a side slope or through cracks in the cap. This is a common issue at landfill sites and highly variable emissions from these hotspots have been reported (Di Trapani et al., 2013; Rachor et al., 2013; Gonzalez-Valencia et al., 2016).

Although they contribute to the total GHG emissions of the landfill, measurements within the close proximity of those hotspots are not suitable for estimation of emissions from the active site. High temporal variability and spatial inhomogeneity would result in non representative fluxes. Hence, the application of the CFD model to the ridge measurements is not presented here. Emissions derived from measurements in greater distance to these hotspots can include their contribution into bulk emission estimates (see section 3.4).

**3.1 Emission ratios**

The ratio of ppm $CH_4$ per ppm $CO_2$ at the location of the emission source is often referred to as emission ratio and is given here in $ppm\,ppm^{-1}$ for simplicity. It can provide insights into the degree of $CH_4$ oxidation at landfill sites (Gebert et al., 2011; Pratt et al., 2013). Under anaerobic conditions the landfill gas is typically enriched in $CH_4$ and results in ratios of 1.2 to 1.5 $ppm\,ppm^{-1}$ for $CH_4$ to $CO_2$ (Lohila et al., 2007; Gebert et al., 2011). On-site continuous monitoring undertaken in a borehole by Ground-Gas Solutions (GGS) detected LFG ranging from 59 to 67 % $CH_4$ and 31 to 42 % $CO_2$, which results in a mean ratio of 1.8 $ppm\,ppm^{-1}$.

As the FTIR is not directly located at the source the observed signals $\chi_{meas}$ of $CH_4$ and $CO_2$ are the combination of the background and the enhanced mixing ratio ($\Delta\chi = \chi_{meas} - \chi_{bg}$) from the active site. From that, the enhancement factor $EF = \Delta CH_4/\Delta CO_2$ is determined (Lefer et al., 1994), which corresponds to the emission ratio as long as there are no additional sources or sinks along the transport pathway. Here we determine the EF directly from the regression slope of $\chi_{CH_4}$ versus to $\chi_{CO_2}$ (Fig. 3) without prior background subtraction, as described in (Yokelson et al., 2013), because background values for $CO_2$ were not available for the whole measurement period. Data for periods influenced by the active site are plotted separately for day (09:00 to 18:00 UTC) and nighttime (21:00 to 06:00 UTC) as the background of $CH_4$ and $CO_2$ is expected to change during the course of a day. That way EF is derived from data with comparable background values. Data inbetween the day and nighttimes showed a gradual shift in background concentration, which leads to artificially lower EF.

[revised manuscript text omitted]

$$\chi_{FTIR,i} = \frac{f_{Source} \cdot F_{\mathrm{CH_4}}}{A} \cdot \chi_{Source,i} \tag{7}$$

A robust fitting method using an M-estimator to reduce the influence of outliers was also tested, but did not have a significant

5   effect on the results. Hence, only the results from the linear least square fit are reported in the following (Table 3). The standard errors are the fit uncertainty of the coefficient. Inferred fluxes range from 0.66 to 0.92 $\mathrm{mg\,m^{-2}\,s^{-1}}$ during daytime and 1.37 to 1.39 $\mathrm{mg\,m^{-2}\,s^{-1}}$ at night. When all daytime data are fitted together an overall flux of $(0.83 \pm 0.04)\,\mathrm{mg\,m^{-2}\,s^{-1}}$ is obtained. This results in $\mathrm{CH_4}$ emissions of 53.3 $\mathrm{kg\,h^{-1}}$ over the active site.

It should be noted that the CFD model

10    turbulence mixing parameters were optimised to match the bag samples of the tracer release experiment . Hence, the CFD outputs correspond to daytime conditions and fluxes calculated for nighttime need to be used with care, but are included here for completeness. Generally, it can not be predicted how the CFD output would change with decreased turbulence, as it would be the case during night, as it highly depends on the location of the measurement and the meteorological conditions. Higher $\mathrm{CH_4}$ emissions at night could also be explained with a decrease in temperature and a reduced activity of $\mathrm{CH_4}$ oxidising

15   bacteria (Scheutz et al., 2009). A small inverse relationship between temperature and $\mathrm{CH_4}$ emissions at this landfill site was also observed by Riddick et al. (2016). Additionally, the fact that activity on the open site, moving vehicles and deposition of new waste, only takes place during the day could contribute to a diurnal pattern in landfill emissions by introducing oxygen rich air into the surface layer of waste.

Enhancements in $\mathrm{CH_4}$ ($\Delta\mathrm{CH_4}$) simulated from the inferred fluxes (Table 3) are shown in Fig. 7 together with the in situ

20   data. Around 200° the measurements are well represented by the model, but model estimates were found to be lower for other wind directions. This is mainly the case for low wind speeds, where more $\mathrm{CH_4}$ can accumulate, and wind directions further south east.

**3.4   Inclusion of an additional source area**

As described in the previous section, the CFD model results in a steep decline in simulated $\mathrm{CH_4}$ concentration at wind direc-

25   tions of 220° and further west, while measurements are still enhanced. No $\mathrm{CH_4}$ emissions were observed on top of the restored section of the landfill site between the ridge and the portakabin, but emission hotspots were detected on the south side of the ridge above the active site, further referred to as side area (see light pink area in Fig. 8 (a)). Thus, we have included a secondary source area $A_{side}$ in our analysis estimated to be 26,400 $\mathrm{m^2}$. Gaps in the top liner along the side allow for $\mathrm{CH_4}$ to escape underneath a soil cover with some vegetation. These emissions are directly adjacent to the emissions from the active site and

30   are thereby also detected by the FTIR for wind coming from the south to southwest. The emission strength compared to the active site is unknown and can be expected to be highly variable (Rachor et al., 2013). To take these into account, a second

CFD run for the described area as emission source was set up. For a normalised source flux of $f_{source} = 1\,\mathrm{g\,s^{-1}}$ concentration distributions as shown in Fig. 8 (b) are modelled.

Flesch et al. (2009) discussed the requirement of having two sensors in different places for a two source problem. But they also describe the possibility of solving the problem with a single sensor, if the range in meteorological conditions is broad

5   enough. Here, we have only one sensor available, but a range in wind speeds and direction for most days. The modelled concentrations were combined with Eq. (8) to calculate the fluxes from both areas under assumption that the measured $CH_4$ concentration is an accumulated signal of the emissions from the active site and the side.

$$\chi_{FTIR,i} = \frac{A_{active} \cdot \chi_{active,i}}{f_{source}} F_{active,i} + \frac{A_{side} \cdot \chi_{side,i}}{f_{source}} F_{side,i} \qquad (8)$$

Equation (8) was applied in two ways. First, a linear least square fit was applied to the data of each day and night separately

10   for each wind speed. Secondly, all daytime data were fitted with a linear least square together to derive a mean flux. Fluxes from both source areas for each set of data are given in Table 4 together with their fit uncertainty as standard error and the residual standard error. The same robust fitting methods were applied again to take outliers into account. The results were found to be consistent with each other within the fit uncertainty. Hence, only results from the linear least square fit are reported.

The combined fit in cases where data are only available for the lower wind directions, such as for $4\,\mathrm{m\,s^{-1}}$ on 9 and 11 August

15   2014, does not result in realistic coefficients for the fluxes and in conjunction with their large errors can not be considered as representative values. For wind speeds of $10\,\mathrm{m\,s^{-1}}$ only two data points were available, i.e. zero degrees of freedom, and the fit assigned a much higher flux to the side area and only a minor contribution to the active site. Therefore these fits were not further included. A longer measurement period would be of benefit to obtain data of a wider range in meteorological conditions.

Figure 9 shows simulated $\Delta CH_4$  based on fluxes calculated from the separate fits with combined CFD runs in

20   comparison to the measurements. At the peak wind direction both approaches show similar good agreement between the model and the measurements. Measured $CH_4$ concentrations at $220°$ are much better represented by the combined CFD model compared to the model run based on the active site only (Fig. 9). The mean residual standard error (RSE) could be reduced from 0.42 to 0.25 ppm based on equivalent fits from 9 August ($6\,\mathrm{m\,s^{-1}}$), 11 August (6 and $8\,\mathrm{m\,s^{-1}}$), 12 August (6 and 8 m s$^{-1}$) and night of 11 to 12 August 2014 (4 and $6\,\mathrm{m\,s^{-1}}$). The mean fluxes from the same daytime data combined in one fit are

25   $(0.71 \pm 0.05)\,\mathrm{mg\,m^{-2}\,s^{-1}}$ for the active site and $(0.32 \pm 0.08)\,\mathrm{mg\,m^{-2}\,s^{-1}}$ for the side. From this the overall emissions are $76.0\,\mathrm{kg\,h^{-1}}$ over an area of $44{,}223\,\mathrm{m^2}$.

Another reason for a discrepancy between modelled and measured $CH_4$ mole fractions could be the parametrisation for the turbulence in the CFD model. A standard fixed turbulent dispersion parametrisation (Sct = 0.7, see section 2.4.2) was used in OpenFOAM assuming to be the best description of the conditions at the landfill site. Similar parametrisation has been used in

30   previous studies by Jeanjean et al. (2015, 2017) for evaluation of the CFD model. Over the landfill site, the turbulent mixing is likely to be variable with changes in roughness and topography across the site. This would subsequently lead to greater modelling errors. Fluctuations in wind speed and direction can lead to uncertainties in the results, if the aggregation time of the

data is too short. This was addressed by averaging over at least five 3-min data points per bin for calculation of the fluxes (see section 3.3).

**3.5 Comparison to other flux estimations**

Based on the CFD approach considering the active area, a mean daytime $CH_4$ flux of $0.83 \, \mathrm{mg \, m^{-2} \, s^{-1}}$ was calculated, which corresponds to $53.3 \, \mathrm{kg \, h^{-1}}$. Including emissions from the side area results in an overall flux of $76.0 \, \mathrm{kg \, h^{-1}}$ over a total area of $44{,}223 \, \mathrm{m^2}$. $CH_4$ fluxes from the landfill site were also measured by two other groups during the landfill campaign. Estimating the actual emitting area is a difficult task. While our focus was on the open active site, Riddick et al. (2016) included the surrounding area as well. Riddick et al. (2016) used an atmospheric inverse dispersion model to determine fluxes from the off-site $CH_4$ measurements between July and September 2014. They assume emissions to be only from the open site, which they estimate to be approximately $70{,}000 \, \mathrm{m^2}$. With $0.709 \, \mathrm{mg \, m^{-2} \, s^{-1}}$ on average over day and night they observed a $CH_4$ flux in good agreement to the one determined in this work. Based on the larger area the total flux in Riddick et al. (2016) corresponds to $178.7 \, \mathrm{kg \, h^{-1}}$. They report an  uncertainty of 42 % that is similar to our approach. Mønster and Scheutz (2015) applied a dynamic tracer dispersion method to estimate total $CH_4$ emissions from the landfill (total area: $330{,}000 \, \mathrm{m^2}$) between 5 and 12 August 2014. They derived fluxes in the range of 217 to $410 \, \mathrm{kg \, h^{-1}}$ with a standard error of 14 to 42 % from six experiments in this period. $CH_4$ emissions estimated by the landfill site's owner are around 2,230 tonnes in 2014, which corresponds to an annual mean flux of $254.6 \, \mathrm{kg \, h^{-1}}$. This value is calculated from the total $CH_4$ as modelled based on waste input to the site and the LFG consumed by the power plant.

Compared to the other two methods we derived a lower $CH_4$ flux from the landfill site based on the on-site measurements at the portakabin. The approaches of Riddick et al. (2016) and Mønster and Scheutz (2015) aim at quantifying the integrated signal of the whole landfill site, while our CFD approach focussed on emissions from the active site only (and separately the side area). Hence, fluxes obtained by these bulk emission methods are likely to be higher, including emissions from other areas, then the ones derived with the CFD approach. Indications for further emissions from wind directions towards the GUP and the temporarily capped completed cell in the south east were visible in the $CH_4$ distribution measured with the FTIR (Fig. 4), but were not subject of the present study. Definition of the source area is a crucial part for setting up the CFD simulation and needs to be carefully assessed. When comparing fluxes inferred from different methods the source areas used need to be accounted for. As an advantage of the CFD approach, several source areas with different emission strength can be included.

**4 Summary and conclusions**

We presented a new approach to quantify $CH_4$ emissions from a defined source area at a landfill site. To this end, precise in situ measurements were combined with a CFD model. The CFD model only needs to be run once to cover the whole range in meteorological conditions and can then be applied to a series of continuous in situ measurements. Additionally, meteorological and background measurements are needed for application of the CFD model, which can easily be maintained over extended periods of time. The FTIR measurements can be conducted over a longer period without great effort and can thereby cover a

wide range of various environmental conditions.

Consistent fluxes from the active site were found for three different days with southerly winds transporting air from the source area towards the portakabin. Data from wind directions of 220° were not well reproduced by the CFD outputs for the active site only. Taking emissions from the side area between the active site and the ridge into account improved the agreement between measurements and model in this area. This shows that the emission source in the CFD model needs to be well defined. This is challenging for a heterogeneous terrain like a landfill site, where several sources of $CH_4$ with different emission strength exist. This is where the CFD model demonstrates its strength by including the complex topography of the site. Chamber measurements or an initial walk over survey with a small portable $CH_4$ sensor are valuable tools to characterise different parts of a landfill site and detect emission hotspots which are otherwise easily overseen by point measurements. It was discussed that measurements in the direct proximity of highly variable point sources such as landfill hotspots are not suitable for the approach with the CFD model. But the position of the instrument should be close enough to detect a signal from the source areas from  a range in wind direction in order to separate areas of different emission strength. The presented method could be improved by using multiple, spatially distributed sampling points. This could be achieved for future applications through the use of all four sample inlets of the FTIR to sample alternately from different points along the cross section of the plume. CFD results could be extracted for all sampling points without further modelling effort.

With our approach we estimated $CH_4$ emissions between 53 and 76 $\mathrm{kg\,h^{-1}}$ by the active site and surrounding area, depending on the area taken into account with the CFD model. These values represent only a snapshot of the landfill emissions based on the short measurement period. Longer-term or repeated measurements in different seasons would be needed to investigate emissions under different meteorological conditions and provide a more complete picture. The main contribution to the uncertainty of the derived emissions results from the limitations of the CFD model simulations. Compared to the total emission estimate from the landfill site's owner (254.6 $\mathrm{kg\,h^{-1}}$) and the bulk emission approaches by Riddick et al. (2016) (178.7 $\mathrm{kg\,h^{-1}}$) and Mønster and Scheutz (2015) (217 to 410 $\mathrm{kg\,h^{-1}}$), this assigns a smaller contribution to the active site and suggests additional significant $CH_4$ emissions from other parts of the site. Enhanced $\Delta CH_4$ was observed for wind directions further east of the active site (Fig. 4) where the CFD model does not show any contribution from the active site. The presented study shows that the CFD approach can be used to assess the emission strength from a well defined area in a complex terrain with several distinguishable emission sources.

**5  Data availability**

Data are available upon request from the authors.

*Acknowledgements.* We thank NERC for their funding (NE/K002465/1 and NE/K002570/1) as part of the Greenhouse gAs Uk and Global Emissions (GAUGE) project. For technical and logistical support we would like to thank the team of Viridor on site. Thanks to Andrew Brunton and John Naylor from Ground-Gas Solutions for providing data from their survey of the landfill site. We would like to thank David Hodgetts from the School of Earth and Environmental Sciences, The University of Manchester, for providing the LIDAR survey data. From the School of Chemistry at the University of Bristol we would like to thank James C. Matthews, Matthew D. Wright, Damien Martin and Dudley Shallcross for conducting the tracer release experiment.Many thanks to Thorsten Warneke and Hella van Asperen from the Institute of Environmental Physics (IUP), University of Bremen, for their advise on the deployment of the FTIR. We also thank Peter Somkuti for processing the meteorological data. This research used the ALICE High Performance Computing Facility at the University of Leicester.

**Table 1.** EF given as ppm $CH_4$ per ppm $CO_2$ with fit uncertainty and $R^2$ as determined from the slope of the regression from the correlation of $CH_4$ to $CO_2$ measured at the portakabin for day (09:00 to 18:00 UTC) and nighttime (21:00 to 06:00 UTC) separately.

| Date | Day/Night | EF (ppm $ppm^{-1}$) | $R^2$ |
|------|-----------|---------------------|-------|
| 09/08 | Day | $0.266 \pm 0.026$ | 0.393 |
| 11/08 | Day | $0.235 \pm 0.012$ | 0.572 |
| 12/08 | Day | $0.163 \pm 0.015$ | 0.499 |
| 09 to 10/08 | Night | $0.241 \pm 0.007$ | 0.857 |
| 11 to 12/08 | Night | $0.234 \pm 0.007$ | 0.655 |

**Table 2.** Mean $CH_4$ fluxes and standard deviations for the ensemble of derived fluxes for each day/night calculated from the binned FTIR data with the CFD results and the respective background values (BG). The uncertainty for each calculated flux value is estimated from error propagation based on the standard deviation of $\Delta CH_4$ per bin and the model uncertainty (see main text for detail). The range of these uncertainties for each day/night is given in the last column.

| Date | Day/Night | BG (ppm) | Flux (mg m$^{-2}$ s$^{-1}$) | Uncert. Flux (%) |
|------|-----------|----------|------------------|------------------|
| **09/08** | Day | 1.898 | $0.99 \pm 0.39$ | 40.4 - 44.9 |
| **11/08** | Day | 1.869 | $0.79 \pm 0.12$ | 40.6 - 43.2 |
| **12/08** | Day | 1.867 | $0.78 \pm 0.11$ | 40.6 - 41.9 |
| **11 to 12/08** | Night | 1.911 | $1.38 \pm 0.26$ | 41.8 - 43.6 |

**Table 3.** Results of a linear least square fit of the CFD model to the in situ data. $CH_4$ fluxes were fitted for each day/night and wind speeds separately. The standard error for the flux, adjusted $R^2$, the residual standard error (RSE) and degrees of freedom (df) are also shown.

| Date | WS (m s$^{-1}$) | CH$_4$ Flux (mg m$^{-2}$ s$^{-1}$) | Stand. Error (mg m$^{-2}$ s$^{-1}$) | Adj. R$^2$ | RSE (ppm) | df |
|---|---|---|---|---|---|---|
| **Day 09/08** | 4 | 0.89 | 0.22 | 0.805 | 0.71 | 3 |
| | 6 | 0.92 | 0.11 | 0.928 | 0.36 | 4 |
| | 8 | 0.80 | | | | 0 |
| **Day 11/08** | 4 | 0.80 | 0.05 | 0.987 | 0.16 | 2 |
| | 6 | 0.87 | 0.10 | 0.950 | 0.27 | 3 |
| | 8 | 0.79 | 0.15 | 0.845 | 0.36 | 4 |
| | 10 | 0.66 | 0.01 | 0.999 | 0.02 | 1 |
| **Day 12/08** | 6 | 0.68 | 0.09 | 0.950 | 0.21 | 2 |
| | 8 | 0.80 | 0.20 | 0.833 | 0.41 | 2 |
| | 10 | 0.90 | 0.02 | 0.999 | 0.04 | 1 |
| **Night 11 to 12/08** | 4 | 1.37 | 0.16 | 0.936 | 0.58 | 4 |
| | 6 | 1.39 | 0.27 | 0.865 | 0.75 | 3 |

**Table 4.** Results of a linear least square fit from the combined CFD model outputs, for the active site and the side, to the in situ data. $CH_4$ fluxes were fitted for each day/night and wind speeds separately. The standard error for the flux, adjusted $R^2$, the residual standard error (RSE) and degrees of freedom (df) are also shown.

| Date | WS | Active site CH$_4$ Flux | Stand. Error | Side CH$_4$ Flux | Stand. Error | Adj. R$^2$ | RSE | df |
|------|----|------|------|------|------|------|------|----|
| | (m s$^{-1}$) | (mg m$^{-2}$ s$^{-1}$) | | (mg m$^{-2}$ s$^{-1}$) | | | (ppm) | |
| **Day 09/08** | 6 | 0.84 | 0.16 | 0.22 | 0.30 | 0.919 | 0.38 | 3 |
| **Day 11/08** | 6 | 0.76 | 0.10 | 0.29 | 0.17 | 0.970 | 0.21 | 2 |
| | 8 | 0.65 | 0.13 | 0.36 | 0.17 | 0.919 | 0.26 | 3 |
| **Day 12/08** | 6 | 0.59 | 0.01 | 0.23 | 0.02 | 1.000 | 0.02 | 1 |
| | 8 | 0.60 | 0.04 | 0.56 | 0.06 | 0.996 | 0.07 | 1 |
| **Night 11 to 12/08** | 4 | 1.23 | 0.21 | 0.36 | 0.35 | 0.936 | 0.58 | 3 |
| | 6 | 1.03 | 0.12 | 0.97 | 0.19 | 0.985 | 0.25 | 2 |

[Figure]

**Figure 1.** Birds view of the landfill site with the active site coloured in red in the centre. The portakabin with the FTIR is located at the north edge of the landfill site. Additional instrumentation was located at the ridge above the active site. A GC used for background measurements was situated about 700 m SW off-site at Inghams Farm and a CRDS was operated on Chalk Hill Lane about 300 m NNE. The entry to the site with the weighbridge and the gas utilisation plant are at the east side.

[Figure]

**Figure 2.** Time series of wind speed (WS, grey) and direction (WD, dark blue) in the top panel and of $CO_2$ and $CH_4$ colour coded with the wind direction. Black and grey refer to background air ($270°$ to $90°$), orange and yellow indicate air coming from the active site and blue to light pink and green colours mark transitional periods.

[Figure]

**Figure 3.** Determination of the enhancement factor as the  $\chi_{CH_4}$  $\chi_{CO_2}$  separately for three days (09:00 to 18:00 UTC) and two nights (21:00 to 06:00 UTC) influenced by air from the active site. Data are shown in two separate panels to account for the different scales.

[Figure]

**Figure 4.** Distribution of $\Delta CH_4$ with wind direction and colour coded with the wind speed based on 15 min averages. The wind direction range of the active site is marked in grey.

[Figure]

**Figure 5.** The emission area used for the CFD approach is marked in red on the topographic map embedded in the British National Grid coordinate system(a). The results of the CFD model for the position of the FTIR measurement site are shown in (b).

[Figure]

**Figure 6.** $\Delta CH_4$ averaged bin wise matching the CFD outputs for each day ((a) - (c)) and the one night (d) with wind coming from the active site. The standard deviation is plotted as error bars. Data are only shown for more then five data points per bin.

[Figure]

**Figure 7.** Measured (MMT) and simulated (CFD fit) $\Delta CH_4$ based on linear fit of the CFD model to the FTIR data.

[Figure]

**Figure 8.** (a) Secondary source area (light pink) between the active site (red) and the ridge and (b) CFD modelled concentration for the location of the FTIR measurements at the portakabin based on the secondary source area only.

[Figure]

**Figure 9.** Measured (MMT) and simulated (CFD combined) $\Delta CH_4$ based on a linear fit combining the CFD model for the active site and the side.